# An Analysis of Factors Influencing Chinese University Students' Major Choice from the Perspective of Gender Differences

Chang Xu [1], Futao Xiang [1], Ruiqi Duan [1], Cristina Miralles-Cardona [2], Xinxin Huo [1] and Junwei Xu [1,*]

1   School of Finance and Public Administration, Anhui University of Finance and Economics, Bengbu 233030, China; xuchang@aufe.edu.cn (C.X.); xiangfutao@aufe.edu.cn (F.X.); duanruiqi@aufe.edu.cn (R.D.); 18119932311@163.com (X.H.)
2   Faculty of Education, University of Alicante, 03690 Alicante, Spain; cristina.miralles@ua.es
*   Correspondence: xujunwei@aufe.edu.cn

**Abstract:** Higher education attainment is a focus of gender differences research. However, little is known about differences in university students' major choices at the same level of higher education in China's education hierarchy. Based on a web survey of 1164 undergraduate students in China's broad enrollment context, this study uses Wenjuanxing to collect information by posting questionnaires on social media platforms and analyzes the impact of gender differences on the major choices of finance and economics undergraduates by using the Linear Discriminant Model (LDM). Moreover, this study explores the differential impact of income level, urban–rural settings, and regional differences on university students' major choices. This study finds that female students are approximately 2.62 times more likely than male students to choose applied majors (such as accounting, financial management, auditing, international business, and so on), a gap that is more pronounced in high-income families and Eastern regions. In addition, in rural areas, female students have a higher probability of choosing applied majors than male students. These findings indicate that in China's broad discipline enrollment model, gender differences still significantly affect students' major choices, and female students are more likely to enter applied majors than their male counterparts.

**Keywords:** broad discipline enrollment system; major choices; undergraduate; gender differences; finance and economics students; China

## 1. Introduction

Since the Reform and Opening-up, China's higher education has expanded rapidly [1,2]. This reform included economic and political changes that influenced the direction and pattern of China's development [3,4]. The original intention of Reform and Opening-up was to achieve the goal of modernization by breaking the closed economic and political system and opening up to the outside world [5]. Under this opening up, China's higher education has experienced unprecedented rapid development [6]. Since then, according to data released by the National Bureau of Statistics, the number of ordinary universities increased from 1022 in 1999 to 2529 in 2022, and the number of students enrolled in these universities incremented from 1.08 million to 7.21 million [7]. Some studies have found that changes in educational stratification and educational expansion are beneficial to women both in China and internationally [6,8–10]. Evidence suggests that the gender gap in education disappeared in the urban labor market in China as early as 2001 [11]. Additionally, since 2005, women have had an advantage over men in university enrollment rates [12]. In 2009, the proportion of female university students in China exceeded that of male students for the first time, reaching 50.48%, and by 2013, the proportion of female university students had reached 51.75%, with 532,000 more female students than male students enrolled at university level [13].

In recent years, with the increasing number of women receiving higher education, gender differences in the quality and structural characteristics of higher education have

become a discussion focus of academic interest [14–16]. According to a survey, in 2013, among the top ten majors with the highest proportion of male students in Chinese universities, all were in Science, Technology, Engineering, or Mathematics (STEM) fields, and male students accounted for over 80% in these fields. Female students were mainly concentrated in majors such as law, literature, economics, and management, with nursing having the highest proportion of female students at 92% [17]. Similar gender patterns in major choices have been observed in other countries such as the United States, Australia, Japan, and Korea, where male students tend to concentrate heavily on STEM fields while female students tend to focus on non-STEM fields such as humanities [18–20]. Due to the higher proportion of men choosing STEM majors, this difference in major distribution may result in greater opportunities for men to enter higher-status occupations, while women are more likely to enter lower-status occupations [21,22].

Scholars argue that societal expectations of different roles and expectations for males and females in specific industries may influence the major choices of male and female students [23]. Engineering, computer science, and other STEM fields are considered more suitable for males, while fields such as education, sociology, and economics are considered more suitable for females. These notions to some extent influence students' level of interest and willingness to pursue certain majors [23]. Additionally, educational background and family environment can also have an impact on the major choices of male and female students [24]. Schools and families often convey specific gender role ideologies through subtle guidance and education, which in turn affect students' major choices [24]. For instance, parents' educational levels serve as a factor; when parents have lower levels of education, they may not consider all aspects when assisting their children in selecting a major [25]. Conversely, when parents have higher levels of education, they are often able to provide valuable advice by analyzing factors such as academic difficulty, major prospects, salary potential, and societal value when guiding their children in choosing a major [25].

Gender differences in major choices may also exacerbate gender differences in the labor market [26–28]. Men and women accumulate different types of professional human capital due to their different university majors, and the job opportunities and salary returns for different majors are directly related to the employment prospects or market demand for these majors [29]. Hence, the gender composition of occupations has strong explanatory power in the gender income differences in the labor market worldwide [16]. Gender differences in the labor market may also further exacerbate people's stereotypes of gender differences in certain majors [30,31]. For example, if one thinks that men are better equipped to do certain types of work, they may prefer male students under similar conditions, especially when selecting STEM students. Over time, this may lead to a shortage of women in certain fields and make it difficult for them to display their talents [32]. Therefore, studying gender differences in undergraduate major choices is not only conducive to formulating more targeted policies for the balanced development of disciplines but also of great significance for scientifically guiding students to choose schools and solving the problem of sustainability of major-career matching.

Unlike existing studies on gender differences in major choices in China [13,16], this paper investigates the impact of gender differences on major choices under the broad discipline enrollment model. In 2014, the Chinese State Council implemented the 'Opinions on Deepening the Reform of the Examination and Enrollment System' [33], and an increasing number of universities began to practice a talent cultivation mode of enrollment and training based on broad disciplines. As of 2021, 117 out of 137 'Double First-Class' (the 'Double First-class' initiative is a national project aimed at building world-class universities and world-class disciplines. It was launched by the Chinese government in 2015 and covers all universities in China that are involved in higher education. The universities that are awarded the 'Double First-Class' status are selected based on a comprehensive evaluation system that ranks them according to their level of academic excellence and research performance. The universities that receive the status are provided with funding and other resources to further support their goal of becoming world-class institutions) universities in

China, accounting for 85.4% of the total, have implemented broad discipline enrollment and training. Similar to some European and American countries, under the broad discipline enrollment system, students only need to choose the broad discipline, such as liberal arts or science, before admission and do not choose a specific major. After 1–2 years of general education courses, students choose their majors based on the streamlining plan.

In the traditional enrollment mode (directly choosing majors after the college entrance examination), students' major choice decisions are mainly influenced by whether they can be admitted to their preferred schools, with specific major choice decisions heavily impacted by their parents. However, under the broad discipline enrollment system, since the choice of major comes after admission, and university students are away from their parents, there is no risk of rejection, which reduces the influence of family, and strengthens students' voluntary choices [25,34]. Existing research mainly focuses on major choices under the traditional enrollment mode in Chinese higher education; moreover, the focus is predominantly on comprehensive universities, but there is little knowledge available on how gender affects major choices during the undergraduate period under broad discipline enrollment. In regard to financial and economic institutions of higher learning, the available information is even scarcer. In the context of broad discipline enrollment in China, how gender affects the decisions of students' major choices remains an unresolved issue.

In order to advance existing research, this study aimed to investigate the relationship between gender differences and undergraduate major choice in a broad-based admissions process of finance and economics university students by empirically testing the influence of gender on students' choices. Further, specifically, this study sought to answer the following three research questions (RQ):

RQ1: Under the current broad discipline enrollment system in China, will gender differences affect major choices in finance and economics?
RQ2: To what extent does gender influence the profession relative to other factors influencing the choice of major?
RQ3: Will the impact of gender differences on professional choices vary depending on family background?

To achieve the research objectives of this study, we rely on survey data from 1164 undergraduate students majoring in universities of finance and economics. Utilizing the linear discriminant model (LDM), we empirically examine the influence of gender differences on major choices. Previous studies using narrative research methods were only able to compare mean differences in major choices between two or more gender groups at a holistic level. The LDM model, by considering the relationships and interactions among multiple variables, enables a quick and accurate estimation of the extent to which gender affects major choices [35].

## 2. Data and Methods

### 2.1. Research Design

This study employed a quantitative research design with the aim of exploring gender differences in major choice. A large-scale representative sample was utilized, and data were collected through structured questionnaires obtained via social media platforms. The questionnaires included items related to demographic information, major aspirations, factors influencing major choices, and perceptions of gender-related barriers. Data analysis primarily utilized LDM to summarize and examine gender differences in major preferences, motivations, and cognitive barriers. The study adhered to ethical guidelines and ensured participant confidentiality and anonymity.

### 2.2. Participants

The participants in this study are undergraduate students studying at universities of finance and economics. There are over 50 finance and economics universities in China, with a total of over 1 million undergraduate students [7]. To ensure a comprehensive representation of universities with varying academic reputations and geographical locations

throughout the country, we extensively invited students from finance and economics universities situated in the Eastern, Central, and Western regions of China to participate in an online survey. The survey was distributed to a total of 1200 students. Participating universities included Shandong University of Finance and Economics, Anhui University of Finance and Economics, and Jiangxi University of Finance and Economics, among others.

The research data was collected via an online survey, ensuring the anonymity of the student participants. The online survey platform Wenjuanxing (https://www.wjx.cn, accessed on 15 July 2023) was utilized for data collection. The survey was formulated and disseminated via hyperlinks on popular Chinese social media platforms, namely Weibo, WeChat, and QQ. The dissemination of these links occurred with the assistance of teachers and classmates who subsequently forwarded them. While our survey methodology was not limited to traditional sampling techniques, we made efforts to maximize participant recruitment through diverse distribution channels to ensure sample diversity and representativeness.

*2.3. Measures*

The dependent variable in our study was the decision of undergraduate students to enroll in an applied or non-applied major at universities of finance and economics. This variable was defined as a binary variable, where a value of 1 indicated a student choosing an applied major and a value of 0 indicated a student not choosing an applied major. Based on the definition of Beecher et al. [36], we consider majors that are vocational, technical, and involve specific scenarios as applied majors. Applied disciplines are characterized by strong practicality and skill requirements, higher alignment with the market, better fitting of enterprise skill requirements, adaptability to complex work needs, and more lucrative returns in the labor market [37]. These majors include Accounting, Financial Management, Auditing, International Business, Marketing, Human Resource Management, Logistics Management, E-commerce, Engineering Cost, Computer Science and Technology, Business English, and others. Non-applied majors include Economics, National Economic Management, Public Finance, Journalism, Statistics, Taxation, Trade Economics, Finance, Business Administration, Mathematics, and others. From the survey results, it was found that students choosing non-applied majors slightly outnumbered those choosing applied majors. Of the 1162 valid questionnaires, 538 students (46.22%) chose an applied major, while 626 students (53.78%) chose a non-applied major.

Because other factors may affect the choice of undergraduate majors, according to Ding et al. [38], we chose as explanatory or independent variables the personal and family characteristics of undergraduate students. These variables include age, academic performance, family income, household registration, and hometown. We measured academic performance using the scores of Calculus and College English Test 4 (CET-4). The average score for Calculus was 71.21, and the score for the CET-4 was 428.66. In terms of family income, 329 respondents (28.26% of total respondents) had a family income below 150,000 yuan, while 185 respondents (15.89% of total respondents) had a family income above 450,000 yuan. In terms of household registration, students with urban registration slightly outnumbered those with rural registration. There were 672 students (57.73% of total respondents) with urban registration and 492 students (42.27% of total respondents) with rural registration. Table 1 presents a synthesis of all variables involved in this study and its codification and interpretation.

In our sample, the number of female students (*n* = 590) was slightly higher than that of male students (*n* = 574). In terms of age, there was little difference between males and females, with an average age of 19.20 years old (the standard deviation is 0.85). Students ranged from freshman to senior, and respondents ranged in age from 17 to 27 years old. Among female students, 282 (49.13% of total female students) were from rural areas, while among male students, 390 (66.10% of total male students) were from rural areas. In terms of academic performance, female students performed better than male students, with an average score of 74.00 in Calculus, which was higher than the male students' average score

of 69.00. The same was true for the CET-4 scores, with female students averaging about 9 points higher than male students. In terms of family income, male students generally had higher family incomes than female students. Only 54 female students (9.15%) were from high-income families, while 131 male students (22.82%) were from high-income families. On the other hand, 222 female students (37.63%) were from low-income families, while only 107 male students (18.64%) were from low-income families. As a whole, 46.48% of the respondents indicated that they came from the Eastern region, 34.71% from the Central region, and 18.81% from the Western region.

**Table 1.** Variable definitions.

| Type | Name | Code | Variable Interpretation | Mean | Standard Deviation |
|---|---|---|---|---|---|
| Dependent variable | Application-oriented specialty | Y | Whether to choose an applied major (1 = Yes; 0 = No) | 0.462 | 0.499 |
| Explanatory variables | Gender | Sex | Gender (1 = male; 0 = female) | 0.493 | 0.500 |
| | Age | Age | Chronological age | 19.199 | 0.850 |
| | Math scores | Math grades | Calculus grades | 71.211 | 11.312 |
| | English scores | English grades | CET-4 scores | 428.659 | 37.009 |
| | Total household income | Income | Annual household income level (1 = less than 150,000; 2 = 15–450,000; 3 = 450,000 or more) | 1.876 | 0.653 |
| | Account type | Rural | Hukou type (1 = rural hukou; 0 = urban hukou) | 0.577 | 0.494 |
| | Hometown | Area | Area where the hometown is located (1 = West; 2 = Central; 3 = East) | 2.277 | 0.760 |

*2.4. Procedure*

Data collection was conducted from January 2022 to March 2023. We mainly based the pre-survey stage on 117 respondents at Anhui University of Finance and Economics (respondents from freshman to senior year, covering all majors in the university, the actual survey of 120 people; 3 people were eliminated because they did not complete all the interviews), and with the permission of the interviewees, we conducted face-to-face interviews with the interviewees, and the main content of the interviews was to clarify the rationale, feasibility, and use of the data collected later. Based on a field survey of 117 participants, we took into account various factors such as student grade level, school differences, household registration, and geographical differences in student origin. In response to the feedback from this pre-survey, we optimized and adjusted the questions in our questionnaire to avoid problems in the pre-survey, and we improved the questionnaire method by improving the way we asked questions and the design of the options. These changes were made to make it easier for attendees to complete questions efficiently and accurately.

Upon gathering a total of 1200 responses, the survey was concluded, ultimately obtaining 1164 valid questionnaires (the recovery rate of valid questionnaires reached 97%; 36 of the questionnaires were missing questions and were not complete, so these 36 questionnaires were excluded). The average completion time for the entire questionnaire was approximately 6 min.

The questionnaire consisted of two main parts. The first part included demographic information about the respondents, such as age, gender, grade level, hometown, and university. The second part contained information on major choice, family background, and other potential factors that could influence major choice. The questionnaire can be obtained by contacting the corresponding author.

### 2.5. Statistical Analyses

The linear probability model (LPM) and the logit model can be used to predict the probability value $P(y \mid x)$ of an event. However, scholars have engaged in heated discussions on which model is superior [39,40]. Linear probability models are based on the assumption of a linear relationship and are estimated using linear regression methods [41]. The logit model is a statistical method used to build binary classification models. It models the relationship between the independent and dependent variables as a logistic function that is used to predict the probability of the dependent variable [41]. In logistic regression, the dependent variable is usually represented as a binomial distribution representing two discrete possible outcomes, and the independent variable is either continuous or discrete. Some scholars argue that LPM is superior to logit because OLS regression is faster than logistic regression, probability changes are more intuitive than changes in odds ratios $(P(y \mid x)/1 - P(y \mid x))$, and the meaning of parameters is more intuitive when interpreting the model [42]. Moreover, due to the problem of quasi-complete separation in the maximum likelihood estimation (MLE) used by the logit model, the logit model may collapse, but the LPM model does not face this issue [43]. However, scholars who support the logit model believe that LPM may produce invalid probability prediction values, while logit does not, and the logit model is less affected by the interaction between explanatory variables, making the parameters more stable. The linear discriminant model (LDM) designed by Allison [35] combines the advantages of LPM and logit. The LDM model is derived from the conversion of LPM and logit models. Firstly, LDM can be converted into a logit model, and secondly, the parameter estimation of LPM can be converted into the maximum likelihood estimation of LDM parameters. Therefore, effective prediction probabilities can be obtained by inserting the converted parameters into the logit model.

Suppose there exists a training set of $n$ independent observations $(x_i, y_i)$, $i = 1, 2,\ldots, n$, such that for any pair $(x, y)$ with given $Y = j$ ($j = 1, 2,\ldots, k$), the distribution of $X$ follows $X{\sim}N$ $(m_j, \sigma)$. Let $n_j$ be the number of observed values in the cluster $Y = j$. In the binary case, where $k = 2$, we estimate the slope $b$ and intercept $a$ in the linear probability model (LPM) using OLS, as described by Allison [35] (Equation (1)).

$$y_i = \alpha + b'x_i + \varepsilon_i \tag{1}$$

By following the parameter transformation derived by Haggstorm [44], we can obtain the estimates $\alpha$ and $\beta$ for the Logit model.

$$\text{Logit}[P(Y = 1|x)] = \ln\left[\frac{P(Y = 1|x)}{P(Y = 0|x)}\right] = \beta x + \alpha \tag{2}$$

In the specific data regression analysis, we applied the Stata17.0 regression software for regression analysis.

## 3. Results

### 3.1. Will Gender Differences Affect Major Choice?

Based on correlation analysis, we fitted the relationship between gender and major selection among undergraduate students, as shown in Figure 1. In our sample, 344 female students (58.31% of total female students) chose applied majors, while 194 male students (33.79% of total male students) chose applied majors. As can be seen in Figure 1, gender influences the major choice differently among undergraduate students. The preliminary fitting results show that the probability of female students choosing applied majors is significantly higher than that of male students (the slope of the fitted line is between −0.64 and −0.54, $R^2 = 0.34$). This result preliminarily indicates that there are gender differences in undergraduate students' professional choices. Next, considering the differences in family background and academic performance between male and female students, we conducted a more rigorous econometric analysis.

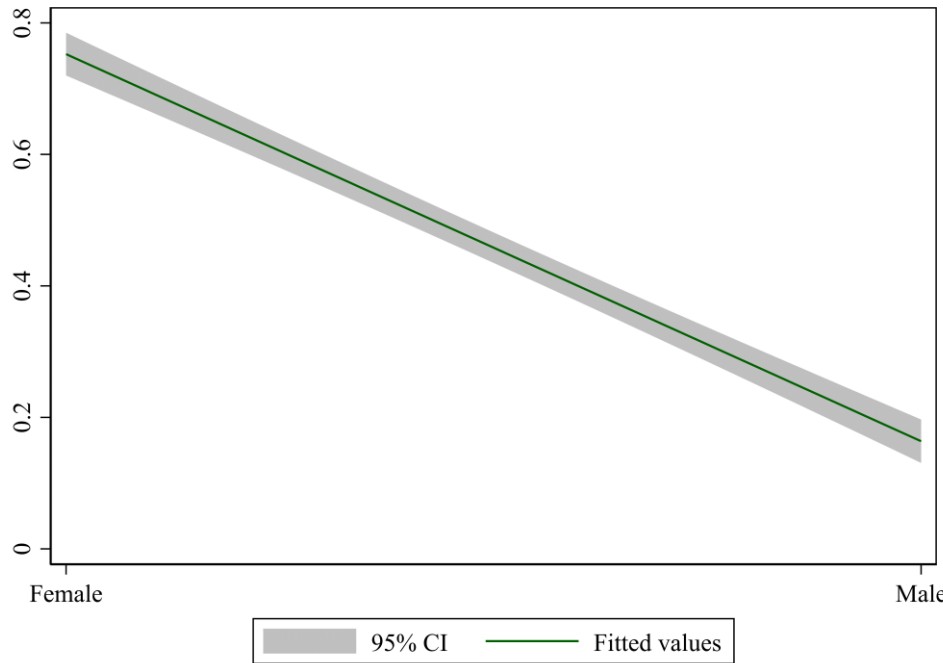

**Figure 1.** Correlation between students' family background and students' major choices. Note: the *x*-axis represents gender and the *y*-axis represents the degree to which a student's family background is correlated with their major choices.

### 3.2. To What Extent Does Gender Impact Major Choice?

Table 2 presents the regression results of the LDM model. Prior to the regression, we conducted a series of checks on the sample. Firstly, we tested the correlation between variables, and the mean VIF (variance inflation factor) was 1.37, with a maximum of 1.99. The test results indicated that there was no serious problem of multicollinearity in the model. Secondly, the Log likelihood of each model ranged from 512.26 to 586.04, and the Pseudo R2 ranged from 0.27 to 0.36, all within reasonable ranges.

Column (1) in Table 2 presents the influence of gender on major choices among undergraduate students. The results show that female students are approximately 2.77 times more likely than male students to choose applied majors, the result being statistically significant at the 1% level ($p < 0.01$). However, gender's impact on major choices may be influenced by individual and family characteristics of undergraduate students. The results in columns (2) and (3) indicate that the regression coefficient of gender decreases after controlling for individual and family characteristics, but still passes the significance test at the 1% level. Column (4) shows the results after adding all the control variables. It can be seen that, with other factors held constant, female students are still about 2.62 times more likely than male students to choose applied majors, and this effect is statistically significant and different from zero.

Regarding the control variables, it can be seen that, with other conditions remaining unchanged, students with better math scores are more likely to choose an applied major. This is related to the transfer policy of universities. Some universities use grades as a threshold for transferring to popular majors, and only students with excellent grades can enter those majors [45]. In terms of income level, students from families with lower income levels are more likely to choose applied majors. From the regression results for hometowns, undergraduates from Western regions are more likely to choose applied majors than those from Eastern and Central regions. This is consistent with the research results on the impact of family income on major choices [16], which shows that students from economically underdeveloped areas often prioritize employment when selecting a major.

**Table 2.** Baseline regression results.

| | (1) | (2) | (3) | (4) |
|---|---|---|---|---|
| Sex | −2.769 *** | −2.744 *** | −2.680 *** | −2.624 *** |
| | (0.151) | (0.159) | (0.160) | (0.167) |
| Age | | −0.079 | | −0.089 |
| | | (0.091) | | (0.092) |
| Math grades | | 0.059 *** | | 0.060 *** |
| | | (0.008) | | (0.008) |
| English grades | | 0.003 | | 0.004 |
| | | (0.002) | | (0.002) |
| Income (medium level vs. low level) | | | −1.109 *** | −1.091 *** |
| | | | (0.178) | (0.183) |
| Income (high level vs. low level) | | | −1.184 *** | −1.327 *** |
| | | | (0.250) | (0.267) |
| Account | | | 0.193 | 0.247 |
| | | | (0.154) | (0.162) |
| Area (East vs. West) | | | −0.911 *** | −0.921 *** |
| | | | (0.220) | (0.228) |
| Area (Central vs. West) | | | −0.399 * | −0.537 ** |
| | | | (0.218) | (0.227) |
| Constant | 1.041 *** | −3.020 | 2.257 *** | −1.807 |
| | (0.122) | (1.927) | (0.282) | (1.970) |
| LR chi2 | 435.891 | 518.911 | 502.283 | 582.474 |
| Pseudo R2 | 0.271 | 0.323 | 0.313 | 0.363 |
| Observations | 1164 | 1164 | 1164 | 1164 |

Note. The numbers in parentheses are robust standard errors; *, **, and *** indicate statistically significant differences at the 10%, 5%, and 1% levels, respectively. Columns (1)–(4) show the regression results when different control variables are added.

### 3.3. Will the Impact of Gender Differences Vary Depending on Family Background?

Figure 2 shows the impact of gender on choosing applied majors as a function of different family characteristics. According to Figure 2, as the family income level increases, the probability of female students choosing applied majors also increases. Particularly, in high-income families, the probability of female students choosing applied majors is 4.27 times that of male students. This indicated that female students from higher-income families were more inclined to choose applied majors than those from lower-income and medium-income families. In terms of household registration differences, and controlling for other demographic characteristics, female students from rural areas are 2.7 times more likely to choose applied majors than male students, slightly higher than those from urban areas (2.54). In terms of regional differences, female students from economically developed Eastern areas had a higher probability of choosing applied majors than those from Central and Western areas.

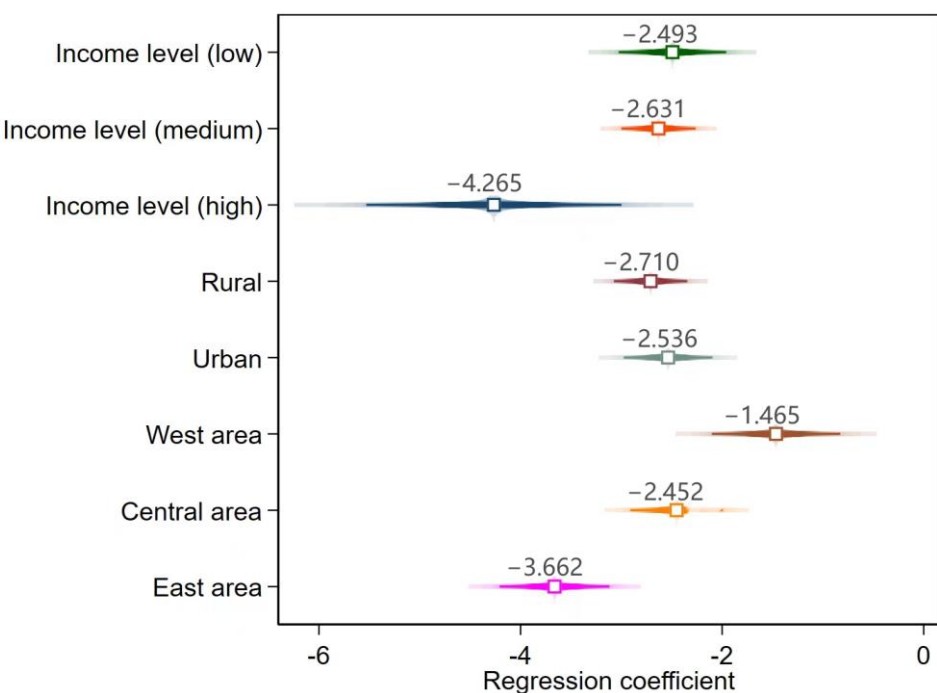

**Figure 2.** The impact of gender differences on professional choice in different family contexts.

## 4. Discussion and Conclusions

### 4.1. Discussion

The analysis allowed the RQs to be answered. The first RQ aims to ascertain the relationship between gender and students' major choices within the context of broad discipline enrollment system backgrounds. Findings indicate significant gender differences in undergraduate students' major choices. This result aligns with the findings of Shao et al. (2022) [23], who used administrative records from a prestigious medical school in China to investigate gender disparities in medical students' specialty choices. Existing research suggests that the disparities in innate capabilities between males and females, arising from biological and genetic factors, such as females being disadvantaged in abstract thinking and scientific cognition, manifest in the selection of disciplines and majors as students enter higher education. Consequently, gender differences in discipline and major choice are exogenous, objective, and unavoidably reflect pre-existing natural endowments [46]. This viewpoint has been extensively corroborated in research examining gender's impact on students' decisions to pursue science, technology, engineering, and mathematics (STEM) disciplines [22]. This study expands upon the existing literature by discussing gender disparities in major choices within the domain of universities of finance and economics.

Regarding the second RQs, the analysis confirms that the probability of female students choosing applied majors is significantly higher than that of male students, approximately 2.77 times higher. This finding is surprising as it appears to contradict some previous studies that reported no advantage for female students in entering popular advantageous majors [47]. Some studies suggest that differences in academic performance may be an important factor contributing to gender disparities in major choices [48]. In this study, even after controlling for academic performance and other potentially influencing variables, the analysis results robustly support the significant tendency of female students to choose popular applied majors over male students. Shao et al. pointed out various obstacles hindering female students from choosing high-paying surgical specialties when analyzing the likelihood of female students selecting such specialties. These obstacles include excessive physical demands, discrimination in recruitment, unfriendly work environments for women, and difficulties in balancing family responsibilities [48]. From the perspective of maintaining social gender segregation, both males and females, influenced by external

factors and driven by internalized different values and potential social discrimination, make distinct decisions in education and major choices, tending towards the professions society expects of them [49]. In the current study, as the majors in finance and economics universities mainly comprise non-STEM disciplines such as accounting and finance, these majors are often considered more suitable for female students by society [23]. Therefore, in our research, female students tend to choose popular applied majors.

The last RQ pertains to the moderating effect of family background on the relationship between gender differences and major choices. This study finds that family background significantly influences gender disparities in major choices. Specifically, females from economically developed regions and high-income households have a greater probability of choosing applied majors. A possible explanation is that females from economically developed regions and high-income households have easier access to resources and opportunities related to applied majors, making them more likely to choose these fields. On the other hand, in regions with relatively lower levels of economic development and income, females may face more social, economic, and cultural constraints, leading them to choose different types of majors. Additionally, female students from rural areas are more inclined to choose popular applied majors [50]. This finding is similar to a study by Yang and Sun [51], which showed that students from rural areas are more likely to choose applied majors than students from urban areas with broader employment prospects.

### 4.2. Limitations

Our study is not without limitations. The first limitation is that we only examined students who have already made their major choices and could not fully capture the mobility of students across different majors. Future longitudinal research could be conducted to investigate the causal relationship between gender and major choices and to explore the specific mechanisms driving these gender differences. The second limitation is that our study only focused on majors within financial and economic universities, which may limit the generalizability of our findings. To address this limitation, future research could expand the scope of the study to include comprehensive universities in order to provide a broader base of evidence. Finally, the time period of our study is during the COVID-19 period, and it is too long compared to the time frame of most other studies, data collection may be affected by the pandemic, and students' choice of major may also be due to changes in the format of the curriculum due to the pandemic, such as from offline to online, resulting in changes in students' attitudes towards certain majors. These can lead to less robust regression results. Future research can explore the impact of these factors on undergraduate majors in finance and economics universities without or as little impact from the pandemic.

### 4.3. Practical Implications

Our findings have significant educational implications for promoting gender equality and social justice in China's higher education system, particularly under the new broad discipline enrollment system. Females are more likely to enter popular applied majors, which can provide them with greater opportunities to alleviate their disadvantaged position in the labor market. However, it is important to acknowledge that applied majors in certain fields, such as finance and accounting, also have limitations in their employment direction. Increased numbers of female graduates in applied majors also create a risk of female employees facing intensified competitive pressures in these industries. Therefore, as long as gender stereotypes persist in the job market, gender equity issues such as opportunities and treatment should be highlighted. Meanwhile, interventions should work towards placing great emphasis on guiding students in their major choices, providing them with adequate professional introductions and employment information to reduce students' lack of understanding or blind choice when selecting majors.

### 4.4. Conclusions

The literature on the major choices in higher education in China has mostly focused on the traditional enrollment model; therefore, there is little research analyzing the impact of gender on major choices under the broad discipline enrollment system. Against the backdrop of the emerging and widely promoted broad discipline enrollment reform in China in recent years, our research has advanced the existing literature. Specifically, the purpose of this study was to analyze how gender affects students' choice of applied majors versus non-applied majors. This includes the degree to which gender differences have an impact on professional choices, and whether this impact will vary depending on family background and region. In recent research, gender classifications have become increasingly diverse. Due to our inability to analyze gender diversity, we follow Cui et al. [52] in categorizing gender mainly as male and female. By using the LDM model to analyze who was more likely to enter advantaged applied majors in universities, we focus on gender differences in professional education resources during the undergraduate stage. Furthermore, we explored the differential impact of gender on major choices due to differences in family background (such as income level, urban and rural areas, and regions). Our study provided empirical evidence that, under the broad discipline enrollment system, the probability of female students choosing applied majors is significantly higher than that of male students and this gap increases with the increase in family income levels. Furthermore, female students from Eastern regions and rural areas in China are more likely to choose applied majors than females in other areas.

**Author Contributions:** Conceptualization, C.X. and J.X.; Methodology, F.X.; Investigation, R.D.; Data curation, F.X. and R.D.; Writing—original draft, C.X.; Writing—review & editing, C.M.-C.; Supervision, X.H.; Funding acquisition, J.X. All authors have read and agreed to the published version of the manuscript.

**Funding:** Anhui University of Finance and Economics "Special Teaching and Research Project of Ideological and Political Courses (Key Project)" (Project No.: ACSZJYZD2022002).

**Institutional Review Board Statement:** Not applicable.

**Informed Consent Statement:** Not applicable.

**Data Availability Statement:** The data used to support the findings of this study are available from the corresponding author upon request.

**Acknowledgments:** We sincerely thank María Cristina Cardona-Moltó's contribution for the revision of this manuscript.

**Conflicts of Interest:** The authors declare no conflict of interest.

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
