# Peer review of "An Analysis of Factors Influencing Chinese University Students’ Major Choice from the Perspective of Gender Differences"

_sustainability, doi:10.3390/su151814037_

Round 1

Reviewer 1 Report

Thanks for opportunity to review manuscript entitled ‘‘An analysis of Factors Influencing University Students' Major Choice from the Perspective of Gender Differences’’ for Sustainability journal. Authors of the manuscript investigated Factors Influencing University Students' Major Choice from the Perspective of Gender Differences in Chinese cultural context. The strengths of the manuscript were that this research investigated the association between variables of interest using a sample of Chinese cultural context and generally replicated findings of previous studies in their cultural context. As an multivariate analyst, and researcher and reviewer in this topic, I think that almost all sections need very significant improvements but it is impossible to accept this article in this form.  Because my main philosophy of reviewing a manuscript as reviewer and sometimes an article editor to improve the manuscript and not punishing the authors, I provided very specific and detailed peer review of the manuscript to increase its quality and citation potential. I hope authors of the manuscript may benefit from my review. Necessary and minor revisions reported section by section with the page and line number and when possible with suggestions.  

Necessary Revisions

Title

1. Page 1, Line 2-3: The title of the article did not accurately reflect the study. Specifically, adding article title Chinese may be useful. Thus, title maybe revised.  One revision may be that ‘‘An analysis of Factors Influencing Chinese University Students' Major Choice from the Perspective of Gender Differences’’ or ‘‘actors Influencing Chinese University Students' Major Choice from the Perspective of Gender Differences’’ These are only suggestions to help authors. Authors also can revise the title with sentences that they think more appropriate for their study.

Abstract

2. Page 1, Line 9: Please revise the following ‘‘Higher education attainment is a hot topic in gender differences research. However, little is known about differences in university students' major choices at the same level of higher education in China's education hierarchy.’’ Please remove hot topic and use another word. It is not a suitable for a scientific article.

3. Page 1, Line 11:  Authors need to add some sample demographics to after aim sentence.  

4. Page 1, Line 11:  Authors need to add sample applied majors in paratheses for better understanding.

5. Page 1, Line 18:  Following sentence ‘‘This means that……’’ 5. Please revise following           ‘‘These findings indicate that …..’’

6. Page 1, Line 21-22:  Authors must remove numbers from keywords.

Introduction

7. Page 1, Line 24: Following sentence is very assertive and needs citation. ‘‘Since the Reform and Opening-up, China's higher education has expanded rapidly.’’

8. Page 1, Line 25-27: Following sentence is very assertive and needs citation. Authors should express this sentence in softer language. ‘‘This reform refers to a series of economic and political transformations that China implemented from the late 20th century to the early 21st century that profoundly changed the direction and pattern of China’s development.’’

9. Page 1, Line 28-29: ‘‘The original intention was to achieve the goal of modernization by breaking down the closed economic and political system, and opening the door to the outside world.’’ Subject missing in the sentence. Do authors mean The original intention of Reform and Opening-up? The citation/citations also needed for sentence.

10. Page 1, Line 30-31: The citation/citations needed for following sentence ‘‘Under this opening, China’s higher education has experienced unprecedented rapid development and became increasingly attractive to international students worldwide.’’

11. Page 1, Line 45: Please remove hot topic and use another word. It is not a suitable for a scientific article.  ‘ ‘become a hot topic of…..’’

12. Page 2, Line 47: Please provide long name of STEM in its first use.

13. Page 1, Line 45-46, Page 2, Line 47-50: The citation/citations needed for following sentence ‘‘According to a survey, in 2013, among the top ten majors with the highest proportion of male students in Chinese universities, all  were in STEM fields, and male students accounted for over 80% in these fields. Female students were mainly concentrated in majors such as law, literature, economics, and management, with nursing having the highest proportion of female students at 92%’’

14. Page 2, Line 56: please add also after may in following sentence  ‘ ‘Gender differences in major selection may…’’

15. Page 2, Line 76: I am not able to understand what authors want to mean with Double First Class’1?. I better explanation is required. Adding footnote to text is better choice.

16. Page 3, Line 102: Authors must revise second research question ‘ ‘With Ceteris paribus, to what extent does gender influence major selection?’’ I think Ceteris paribus is not a English word.

17. Page 1, Page 3, Introduction, General: There are three main weakness in Introduction. Firstly, authors must give more information linear discriminant model (LDM) narratively.   

18. Page 1, Page 3, Introduction, General: Second, authors must give information about advantages of examining gender differences using this method instead of used methods in previous studies narratively.

19. Page 1, Page 3, Introduction, General: According to APA 7 rules each paragraph consists of 3 to 8 sentences. Please check Introduction and along the manuscript and correct as suggested and avoid overly long paragraphs.

Method

20. Page 3-Page 5: Method, General: Authors must construct Method section with following subtitles and with the same order. Research design, Participants or Population and Sample, Measures, Procedure, and Statistical Analyses and must move all related information to related sections.

21. Page 3-Page 5: Method, General: Research design section is completely missing and must be added.

22. Page 3, Line 108-109: The citation/citations needed for following sentence ‘ ‘There are over 50  finance and economics universities in China, with a total of over 1 million undergraduate students.’’

23. Page 3, Line 117-119: In the following sentence ‘‘In our sample, the number of female students (n = 590) was slightly higher than that 117 of male students (n = 574). In terms of age, there was little difference between males and 118 females, with an average age of 19 years old (ranging from freshman to senior year, 17-27 years old).’’ Small ns representing subsample size must be italic. Authors must report age with two decimals with its standard deviation. Grade level distribution must be in a separate sentence.

24. Page 3, Line 122-123: In the following sentence ‘ ‘In terms of academic performance, female students performed better than male students, with an average score of 74 in Calculus, which was higher than the male students' average score of 69.’’ Authors must report average scores with two decimals e.g., 69.12.

25. Page 3, Line 126-129: I think authors forget to add percentages following sentences. ‘‘Only 154 female students (add %) were from high-income families, while 131 male students (add %) were from high-income families. On the other hand, 222 female students (add %) were from low-income families, while only 107 male students (add %) were from low-income families.’’

26. Page 3-Page 5: Method, General: Authors must give information about sampling procedure in Participants section (convenience sampling?)

27. Page 3, Page 4 Line 133-142: Well done and written.

28. Page 4 Line 152: What is the reason for removing 36 participants add a reason sentence?

29. Page 4 Line 152:  Following must be at the end of Data collection process or Procedure section. ‘‘Average completion time for the entire questionnaire was approximately 6 minutes.’’

30. Page 4 Line 166:   It is unclear what authors want to mean with micro-level scenarios?

31. Page 4 Line 194:   Following information is completely wrong and must remove ‘ ‘When the dependent variable y is a binary or multinomial discrete variable in a linear regression model’’ true maybe generalized linear model wo of them is completely different things.

32. Page 3-Page 5: Method, General: Authors must add which statistical software used in their analyses.

33. Page 3-Page 5: Method, General: No need to Line 212-219 and must remove from manuscript.

Results

34. Page 6, Page 12, Results section general: Although Sustainability journal instructions did not give authors a systematic reporting for statistical results. In such cases, using a common format is a good choice such as APA 7. Moreover, along the manuscript reporting of statistics need to be completely correct according to APA 7 reporting.

35. Page 6, Page 12, Results section general: Authors must report all findings with the same decimal. Apa recommend two but for more precision three decimal may be reported. Along this section authors must correct reporting of findings with two or three decimal. This comments is valid for tall tables in results section.

36. Page 6, Page 12, Results section general: Under the tables, instead of Authors must use Note. This comment is valid for tall tables in results section.

37. Page 6, Page 12, Results section general: Statistical sybols writings must be corrected along the manuscript.

38. Page 6, Page 12, Results section general: Authors must double check if figure 2 title is correct or not?

39. Page 6-Page 9 Results General: Results section and Discussion must be separate.

40. Page 6-Page 9 Results General: Authors must construct Discussion section with following subtitles Discussion, Limitations, Practical implications, Conclusion and must move all related information to related sections.

Moderate editing of English language required

Author Response

Response to comments by Reviewer #1

Thanks for opportunity to review manuscript entitled ‘‘An analysis of Factors Influencing University Students' Major Choice from the Perspective of Gender Differences’’ for Sustainability journal. Authors of the manuscript investigated Factors Influencing University Students' Major Choice from the Perspective of Gender Differences in Chinese cultural context. The strengths of the manuscript were that this research investigated the association between variables of interest using a sample of Chinese cultural context and generally replicated findings of previous studies in their cultural context. As an multivariate analyst, and researcher and reviewer in this topic, I think that almost all sections need very significant improvements but it is impossible to accept this article in this form.  Because my main philosophy of reviewing a manuscript as reviewer and sometimes an article editor to improve the manuscript and not punishing the authors, I provided very specific and detailed peer review of the manuscript to increase its quality and citation potential. I hope authors of the manuscript may benefit from my review. Necessary and minor revisions reported section by section with the page and line number and when possible with suggestions.  

Response : We appreciate your thorough review and suggestions for improvement. We understand that significant improvements are needed in almost all sections of the manuscript. We value your expertise and guidance in enhancing the quality of our research and will carefully consider your specific recommendations. Thank you once again for taking the time to review our paper. Our responses are listed below. The original comments are printed in black and are followed by our responses in blue.

  1. Page 1, Line 2-3: The title of the article did not accurately reflect the study. Specifically, adding article title Chinese may be useful. Thus, title maybe revised.  One revision may be that ‘‘An analysis of Factors Influencing Chinese University Students' Major Choice from the Perspective of Gender Differences’’ or ‘‘actors Influencing Chinese University Students' Major Choice from the Perspective of Gender Differences’’ These are only suggestions to help authors. Authors also can revise the title with sentences that they think more appropriate for their study.

Response 1: Thank you for your valuable suggestion regarding the title of our manuscript. We appreciate your input and have accepted your recommendation. As a result, we have revised the title to "An analysis of Factors Influencing Chinese University Students' Major Choice from the Perspective of Gender Differences." We believe that this revised title accurately reflects the scope and context of our study. Once again, we sincerely appreciate your guidance and expertise in improving the clarity and relevance of our research.

  1. Page 1, Line 9: Please revise the following ‘‘Higher education attainment is a hot topic in gender differences research. However, little is known about differences in university students' major choices at the same level of higher education in China's education hierarchy.’’ Please remove hot topic and use another word. It is not a suitable for a scientific article.

Response 2: Thank you for your valuable suggestion in regards to the language used in our manuscript. We appreciate your guidance on maintaining a suitable tone for a scientific article. We have carefully considered your recommendation and have made the necessary revisions. We believe that this revision better aligns with the formal and professional tone expected in a scientific article. Once again, we sincerely appreciate your expertise and feedback in improving the quality of our manuscript.

In lines 10 to 12 of the revised manuscript, the main changes are as follows:

“Higher education attainment is a focus in gender differences research. However, little is known about differences in university students' major choices at the same level of higher education in China's education hierarchy.”

  1. Page 1, Line 11:  Authors need to add some sample demographics to after aim sentence.  

Response 3: Thank you for your suggestion to add sample demographics to our manuscript. Thank you for your guidance in providing more background and details relevant to study participants. We believe that this addition will improve the clarity and integrity of our manuscripts. Once again, we sincerely thank you for your expertise and feedback in improving the quality of our research.

Based on your suggestions and those of other reviewers, we have revised the target sentence.

In lines 12 to 16 of the revised manuscript, the main changes are as follows:

“Based on a web survey of 1164 undergraduate students in China's broad enrollment context,

this study uses Wenjuanxing (https://www.wjx.cn) to collect information by posting questionnaires on social media platforms, and analyzes the impact of gender differences on the major choices of finance and economics undergraduates by using the Linear Discriminant Model (LDM).”

  1. Page 1, Line 11:  Authors need to add sample applied majors in paratheses for better understanding.

Response 4: Thank you for your suggestion regarding providing more specific information about the applied majors in our manuscript. We appreciate your guidance in improving the reader's understanding of the study. Based on your recommendation, we have made the following modification in the revised manuscript. We believe that this addition will provide a clearer picture of the specific applied majors analyzed in our study. Once again, we sincerely appreciate your expertise and feedback in improving the quality of our research.

In lines 17 to 20 of the revised manuscript, the main changes are as follows:

“The study finds that female students are approximately 2.62 times more likely than male students to choose applied majors (such as Accounting, Financial Management, Auditing, International Business and so on), a gap that is more pronounced in high-income families and Eastern regions.”

  1. Page 1, Line 18:  Following sentence ‘‘This means that……’’ 5. Please revise following ‘‘These findings indicate that …..’’

Response 5: Thank you for your suggestion regarding the sentence “These findings indicate that” in our manuscript. We appreciate your guidance in improving the clarity and precision of our statement. Based on your recommendation, we have made the following modification in the revised manuscript. We believe that this revision better conveys the implications of the findings in a concise manner. Once again, we sincerely appreciate your expertise and feedback in enhancing the quality of our research.

In lines 21 to 24 of the revised manuscript, the main changes are as follows:

“These findings indicate that China's broad discipline enrollment model, gender differences still significantly affect students' major choices, and female students are more likely to enter applied majors than their male counterparts.”

  1. Page 1, Line 21-22:  Authors must remove numbers from keywords.

Response 6: Thank you for your suggestion regarding removing numbers from the keywords in our manuscript. We appreciate your guidance in aligning the format of the keywords with the standard conventions. Based on your recommendation, we have removed the numbers from the keywords as per the revised manuscript. We believe that this modification adheres to the appropriate formatting guidelines for keywords. Once again, we sincerely appreciate your expertise and feedback in improving the presentation of our research.

In lines 25 to 26 of the revised manuscript, the main changes are as follows:

“Keywords: broad discipline enrollment system; major choices; undergraduate; gender differences; finance and economics students; China”

  1. Page 1, Line 24: Following sentence is very assertive and needs citation. ‘‘Since the Reform and Opening-up, China's higher education has expanded rapidly.’’

Response 7: We appreciate your guidance in ensuring the accuracy and credibility of our statement. Based on your recommendation, we have added a citation to support this claim in the revised manuscript. We believe that this addition strengthens the validity of our statement by providing a credible source to support the assertion. Once again, we sincerely appreciate your expertise and feedback in improving the quality of our research.

In lines 29 to 30 of the revised manuscript, the main changes are as follows:

“Since the Reform and Opening-up, China's higher education has expanded rapidly [1,2].”

  1. W. Cheng, “Chinese Wisdom in the Development of Higher Education,” Higher Education Exploration, 2022, pp. 7-13.
  2. W. Xiong, J. Yang and W. Hen, “Higher education reform in China: A comprehensive review of policymaking, implementation, and outcomes since 1978,” China Economic Review, 2022.
  3. Page 1, Line 25-27: Following sentence is very assertive and needs citation. Authors should express this sentence in softer language. ‘‘This reform refers to a series of economic and political transformations that China implemented from the late 20th century to the early 21st century that profoundly changed the direction and pattern of China’s development.’’

Response 8: Thank you for your suggestion regarding providing a citation for the assertive statement in our manuscript. We appreciate your guidance in ensuring the accuracy and credibility of our writing style. Based on your recommendation, we have added a citation in the revised manuscript to support the claim about the transformative nature of the reforms implemented by China from the late 20th century to the early 21st century. We believe that these modifications enhance the validity and tone of our statement. Once again, we sincerely appreciate your expertise and feedback in improving the quality of our research.

In lines 30 to 31 of the revised manuscript, the main changes are as follows:

“This reform included economic and political changes that influenced the direction and pattern of China's development [3,4].”

  1. X. Zhao, “An Analysis on the Characteristics of lnstitutional Changes of Reform and Opening up and the Transformation of Economic Thoughts Development,” Bulletin of the History of Economic Thought, 2021, pp.25-57.
  2. A. Wang, Y. Yang, S. Zhang, Y. Du and Y. Zeng, “Research on economic development trend of reform and opening up: based on big odellingeling analysis method,” Procedia Computer Science, 2023, pp.533-540.
  3. Page 1, Line 28-29: ‘‘The original intention was to achieve the goal of modernization by breaking down the closed economic and political system, and opening the door to the outside world.’’ Subject missing in the sentence. Do authors mean The original intention of Reform and Opening-up? The citation/citations also needed for sentence.

Response 9: We appreciate your attention to detail in ensuring the clarity and coherence of our writing. Based on your suggestion, we have made the necessary revision to explicitly mention that "The original intention of the Reform and Opening-up" was to achieve the goal of modernization by breaking down the closed economic and political system and opening the door to the outside world. Furthermore, in order to provide proper support for this statement, we have added relevant citations in the revised version of the manuscript. The specific citation will be included to substantiate the original intention and goals of the Reform and Opening-up.

In lines 31 to 33 of the revised manuscript, the main changes are as follows:

“The original intention of Reform and Opening-up was to achieve the goal of modernization by breaking the closed economic and political system and opening up to the outside world [5].”

  1. W. Hu, “On the intrinsic relationship between reform and opening up and socialist modernization,” Scientific Socialism, 2023, pp. 25-31.
  2. Page 1, Line 30-31: The citation/citations needed for following sentence ‘‘Under this opening, China’s higher education has experienced unprecedented rapid development and became increasingly attractive to international students worldwide.’’

Response 10: We appreciate your attention to ensuring the credibility and accuracy of our writing. Based on your feedback and that of other reviewers, we have included the necessary citations and revised this sentence in the revised version of the manuscript to support the statement about China's higher education experiencing unprecedented rapid development. The added citation will provide evidence and support for the mentioned developments in China's higher education system. We sincerely appreciate your careful review and valuable feedback, which have contributed to the improvement of our research.

In lines 33 to 34 of the revised manuscript, the main changes are as follows:

“Under this opening up, China's higher education has experienced unprecedented rapid development [6].”

  1. J. Zhang and Y. Wang, “Achievements of China's Higher Education Since the Reform and Opening up and Their Enlightenment to ‘Double First-Class’ Construction Path,” Journal of Tianjin University (Social Sciences), 2021, pp. 50-57.
  2. Page 1, Line 45: Please remove hot topic and use another word. It is not a suitable for a scientific article. “become a hot topic of…..”

Response 11: Thanks for the suggestion. We appreciate your attention to maintaining a professional tone and style. Based on your recommendation, we have replaced the phrase "become a hot topic of" with the term "discussion focus."  We believe that this modification better aligns with the formal and objective nature of a scientific article. We sincerely appreciate your guidance and valuable feedback, which have contributed to the improvement of the quality of our research.

In lines 46 to 48 of the revised manuscript, the main changes are as follows:

“In recent years, with the increasing number of women receiving higher education, gender differences in the quality and structural characteristics of higher education have become a discussion focus of academic interest [14,15,16].”

  1. M. Pérez-Martín and L. Villardón-Gallego, “University Experiences of Students in a Gender Minority,” Sustainability, 2023, vol. 15. https://doi.org/10.3390/su15054054
  2. D. Sanabrias-Moreno, M. Sánchez-Zafra, M. Zagalaz-Sánchez and J. Cachón-Zagalaz, “Emotional intelligence, quality of life, and concern for gender perspective in future teachers,” Sustainability, 2023, vol. 15. https://doi.org/10.3390/su15043640
  3. C. Fernández-Morante, B.Cebreiro-López, M. Rodríguez-Malmierca and L. Casal-Otero, “Adaptive Learning Supported by Learning Analytics for Student Teachers’ Personalized Training during in-School Practices,” Sustainability, 2022, vol. 14(1). https://doi.org/10.3390/su14010124

12.Page 2, Line 47: Please provide long name of STEM in its first use.

Response 12: Thank you for your feedback regarding the use of acronyms in our manuscript. We appreciate your attention to clarity and consistency in scientific writing.

We added its full name where “STEM” first appeared, and its full name is “Science, Technology, Engineering or Mathematics.” We believe that this adjustment improves the readability and understanding for readers who may not be familiar with the acronym.

We sincerely appreciate your expertise and valuable input, which have contributed to enhancing the quality of our research.

In lines 48 to 51 of the revised manuscript, the main changes are as follows:

“According to a survey, in 2013, among the top ten majors with the highest proportion of male students in Chinese universities, all were in Science, Technology, Engineering or Mathematics (STEM) fields, and male students accounted for over 80% in these fields.”

  1. Page 1, Line 45-46, Page 2, Line 47-50: The citation/citations needed for following sentence ‘‘According to a survey, in 2013, among the top ten majors with the highest proportion of male students in Chinese universities, all were in STEM fields, and male students accounted for over 80% in these fields. Female students were mainly concentrated in majors such as law, literature, economics, and management, with nursing having the highest proportion of female students at 92%’’

Response 13: Thanks for the suggestion. We appreciate your attention to ensuring the accuracy and credibility of our research. Following your recommendation, we have added the necessary citations to support the statement about the distribution of male and female students in different majors in Chinese universities. The specific citation will provide reliable sources of data to validate the mentioned statistics, such as the proportion of male and female students in STEM fields and other majors. We sincerely appreciate your careful review and valuable feedback, which have contributed to improving the quality and reliability of our research.

In lines 48 to 53 of the revised manuscript, the main changes are as follows:

“According to a survey, in 2013, among the top ten majors with the highest proportion of male students in Chinese universities, all were in Science, Technology, Engineering or Mathematics (STEM) fields, and male students accounted for over 80% in these fields. Female students were mainly concentrated in majors such as law, literature, economics, and management, with nursing having the highest proportion of female students at 92% [17].”

  1. A survey report on the employment status of various majors in Chinese universities in 2013, http://bi.hit.edu.cn/index.html.
  2. Page 2, Line 56: please add also after may in following sentence “Gender differences in major selection may…”

Response 14: Thank you for your suggestion regarding the improvement of sentence structure in our manuscript. We appreciate your attention to clarity and coherence in the writing. We have revised the sentence based on your recommendation. This modification helps to enhance the readability and flow of the sentence. We sincerely appreciate your valuable feedback, which has contributed to the overall enhancement of our research.

In lines 75 to 76 of the revised manuscript, the main changes are as follows:

“Gender differences in major choices may also exacerbate gender differences in the labor market [26,27,28].”

  1. A. Fernandes, M. Huber and C. Plaza, “When does gender occupational segregation start? An experimental evaluation of the effects of gender and parental occupation in the apprenticeship labor market,” Economics of Education Review, 2023. https://doi.org/10.1016/j.econedurev.2023.102399
  2. L. Azzollini, R. Breen and B. Nolan, “From gender equality to household earnings equality: The role of women’s labour market outcomes across OECD countries,” Research in Social Stratification and Mobility, 2023. https://doi.org/10.1016/j.rssm.2023.100823
  3. P. Yu and T. Hsieh, “Social stratification in higher education investment: An analysis of students’ choices of college majors and pathways to future labor-market outcomes in Taiwan,” International Journal of Educational Research, 2022. https://doi.org/10.1016/j.ijer.2022.101953
  4. Page 2, Line 76: I am not able to understand what authors want to mean with Double First Class’1 ?. I better explanation is required. Adding footnote to text is better choice.

Response 15: Thank you for your valuable feedback regarding the clarity of the term "Double First Class" in our manuscript. We appreciate your attention to ensuring the thorough understanding of the concepts mentioned. To address this concern, we have made the necessary modifications by adding a footnote to provide a detailed explanation of the term "Double First Class." This clarification aims to provide readers with a better understanding of the concept and its significance. We sincerely appreciate your insightful input, which has contributed to improving the clarity and comprehensibility of our research.

On the third page of the revised manuscript, we added a footnote where "Double First Class" appeared. The footnote reads as follows:

“The ‘Double First-class’ initiative is a national project aimed at building world-class universities and world-class disciplines. It was launched by the Chinese government in 2015 and covers all universities in China that are involved in higher education. The universities that are awarded the ‘Double First-Class’ status are selected based on a comprehensive evaluation system that ranks them according to their level of academic excellence and research performance. The universities that receive the status are provided with funding and other resources to further support their goal of becoming world-class institutions.”

  1. Page 3, Line 102: Authors must revise second research question “With Ceteris paribus, to what extent does gender influence major selection?’’ I think Ceteris paribus is not a English word.

Response 16: Thanks for the suggestion. We appreciate your attention to clarity and appropriate language usage. Upon your suggestion, we have revised the research question to replace “Ceteris paribus” with an English equivalent. The revised research question now reads, “To what extent does gender influence the profession relative to other factors influencing the choice of major?” This modification ensures the use of clear and accessible language for a broader readership. We sincerely appreciate your feedback, which has contributed to improving the quality and readability of our research.

In lines 123 to 124 of the revised manuscript, the main changes are as follows:

“RQ2: To what extent does gender influence the profession relative to other factors influencing the choice of major?”

  1. Page 1, Page 3, Introduction, General: There are three main weakness in Introduction. Firstly, authors must give more information linear discriminant model (LDM) narratively.   

Response 17: Thank you for your valuable feedback regarding the weaknesses in the Introduction section of our manuscript. We appreciate your attention to detail and the improvement of the narrative flow. Considering your suggestion, we provided additional information about the linear discriminant model (LDM) in a more narrative manner. This will help to enhance the reader's understanding of the model and its relevance to our research. We sincerely appreciate your insightful feedback, which will contribute to the overall improvement of our manuscript.

In lines 127 to 134 of the revised manuscript, the main changes are as follows:

“To achieve the research objectives of this study, we rely on survey data from 1164 undergraduate students majoring in universities of finance and economics. Utilizing the Linear Discriminant Model (LDM), we empirically examine the influence of gender differences on major choices. Previous studies using narrative research methods were only able to compare mean differences in major choices between two or more gender groups at a holistic level. The LDM model, by considering the relationships and inter-actions among multiple variables, enables a quick and accurate estimation of the ex-tent to which gender affects major choices [36].”

  1. P. Allison, “Better predicted probabilities from linear probability models with applications to multiple imputation,” Stata Conference Stata Users Group, 2020.
  2. Page 1, Page 3, Introduction, General: Second, authors must give information about advantages of examining gender differences using this method instead of used methods in previous studies narratively.

Response 18: Thanks for the suggestion. We appreciate your suggestion to provide a narrative explanation of the advantages of examining gender differences using the method employed in our study, as opposed to previously used methods. To address this concern, we expanded upon the advantages of our chosen method and provide a narrative explanation of its benefits in comparison to other methods used in previous studies. This additional information will help readers understand the rationale behind our approach and its contributions to the existing literature.

In lines 127 to 134 of the revised manuscript, the main changes are as follows:

“To achieve the research objectives of this study, we rely on survey data from 1164 undergraduate students majoring in universities of finance and economics. Utilizing the Linear Discriminant Model (LDM), we empirically examine the influence of gender differences on major choices. Previous studies using narrative research methods were only able to compare mean differences in major choices between two or more gender groups at a holistic level. The LDM model, by considering the relationships and inter-actions among multiple variables, enables a quick and accurate estimation of the ex-tent to which gender affects major choices [36].”

  1. P. Allison, “Better predicted probabilities from linear probability models with applications to multiple imputation,” Stata Conference Stata Users Group, 2020.
  2. Page 1, Page 3, Introduction, General: According to APA 7 rules each paragraph consists of 3 to 8 sentences. Please check Introduction and along the manuscript and correct as suggested and avoid overly long paragraphs.

Response 19: Thank you for your valuable feedback regarding the paragraph structure in our manuscript's Introduction section. We appreciate your attention to the guidelines provided by APA 7 and the importance of maintaining appropriate paragraph lengths. To address this concern, we will carefully review the Introduction, as well as the rest of the manuscript, and make necessary corrections to ensure that each paragraph consists of 3 to 8 sentences. We will also ensure that overly long paragraphs are appropriately divided to enhance readability. We sincerely appreciate your helpful suggestions, as they will contribute to the overall improvement of our manuscript's structure and presentation.In line 102 of the revised article, we have segmented according to your recommendations and the rules of APA7.

  1. Page 3-Page 5: Method, General: Authors must construct Method section with following subtitles and with the same order. Research design, Participants or Population and Sample, Measures, Procedure, and Statistical Analyses and must move all related information to related sections.

Response 20: Thank you for your valuable feedback on the Method section of our manuscript. We appreciate your suggestion to organize the section using specific subtitles in a consistent order. This will help improve the clarity and structure of our research design. To address this concern, we will revise the Method section, incorporating the suggested subtitles in the following order: Research Design, Participants or Population and Sample, Measures, Procedure, and Statistical Analyses. We have placed all relevant information appropriately under the corresponding subsections. We sincerely appreciate your insightful input, as it will contribute to the overall enhancement of our manuscript's organization and readability.

In lines 136 to 271 of the revised manuscript, the main changes are as follows:

2.1. Research design

This study employed a quantitative research design with the aim of exploring gender differences in major choice. A large-scale representative sample was utilized, and data was collected through structured questionnaires obtained via social media platforms. The questionnaires included items related to demographic information, major aspirations, factors influencing major choice, and perceptions of gender-related barriers. Data analysis primarily utilized LDM to summarize and examine gender differences in major preferences, motivations, and cognitive barriers. The study adhered to ethical guidelines and ensured participant confidentiality and anonymity.

2.2. Participants

The participants in this study are undergraduate students studying at universities of finance and economics. There are over 50 finance and economics universities in China, with a total of over 1 million undergraduate students [7]. To ensure a comprehensive representation of universities with varying academic reputations and geographical locations throughout the country, we extensively invited students from finance and economics universities situated in the Eastern, Central, and Western regions of China to participate in an online survey. The survey was distributed to a total of 1,200 students. Participating universities included Shandong University of Finance and Economics, Anhui University of Finance and Economics and Jiangxi University of Finance and Economics, among others.

The research data was collected via an online survey, ensuring the anonymity of the student participants. The online survey platform Wenjuanxing (https://www.wjx.cn) was utilized for data collection. The survey was formulated and disseminated via hyperlinks on popular Chinese social media platforms, namely Weibo, WeChat, and QQ. The dissemination of these links occurred with the assistance of teachers and classmates who subsequently forwarded them. While our survey methodology was not limited to traditional sampling techniques, we made efforts to maximize participant recruitment through diverse distribution channels to ensure sample diversity and representativeness.

2.3. Measures

The dependent variable in our study was the decision of undergraduate students to enroll in an applied or non-applied major at universities of finance and economics. This variable was defined as a binary variable, where a value of 1 indicated a student choosing an applied major and a value of 0 indicated a student not choosing an applied major. Based on the definition of Beecher et al. [37], we consider majors that are vocational, technical, and involve specific scenarios as applied majors. Applied disciplines are characterized by strong practicality and skill requirements, higher alignment with the market, better fitting of enterprise skill requirements, adaptability to complex work needs, and more lucrative returns in the labor market [38]. These majors include Accounting, Financial Management, Auditing, International Business, Marketing, Human Resource Management, Logistics Management, E-commerce, Engineering Cost, Computer Science and Technology, Business English, and others. Non-applied majors include Economics, National Economic Management, Public Finance, Journalism, Statistics, Taxation, Trade Economics, Finance, Business Administration, Mathematics, and others. From the survey results, it was found that students choosing non-applied majors slightly outnumbered those choosing applied majors. Of the 1,162 valid questionnaires, 538 students (46.22%) chose an applied major, while 626 students (53.78%) chose a non-applied major.

Because other factors may affect the choice of undergraduate majors, according to Ding et al. [39], we chose as explanatory or independent variables the personal and family characteristics of undergraduate students. These variables include age, academic performance, family income, household registration, and hometown. We measured academic performance using the scores of Calculus and College English Test 4 (CET-4). The average score for Calculus was 71.21, and the score for CET-4 was 428.66. In terms of family income, 329 respondents (28.26% of total respondents) had a family income below 150,000 yuan, while 185 respondents (15.89% of total respondents) had a family income above 450,000 yuan. In terms of household registration, students with urban registration slightly outnumbered those with rural registration. There were 672 students (57.73% of total respondents) with urban registration and 492 students (42.27% of total respondents) with rural registration. Table 1 presents a synthesis of all variables involved in this study and its codification and interpretation.

In our sample, the number of female students (n = 590) was slightly higher than that of male students (n = 574). In terms of age, there was little difference between males and females, with an average age of 19.20 years old (The standard deviation of mean age is 0.85). Students ranged from freshman to senior, and respondents ranged in age from 17 to 27 years old. Among female students, 282 (49.13% of total female students) were from rural areas, while among male students, 390 (66.10% of total male students) were from rural areas. In terms of academic performance, female students performed better than male students, with an average score of 74.00 in Calculus, which was higher than the male students' average score of 69.00. The same was true for the CET-4 scores, with female students averaging about 9 points higher than male students. In terms of family income, male students generally had higher family incomes than female students. Only 54 female students (9.15%) were from high-income families, while 131 male students (22.82%) were from high-income families. On the other hand, 222 female students (37.63%) were from low-income families, while only 107 male students (18.64%) were from low-income families. As a whole, 46.48% of the respondents indicated that they came from the Eastern region, 34.71% from the Central region, and 18.81% from the Western region.

Table 1. Variable definitions.

Type

Name

Code

Variable interpretation4

Mean

Standard Deviation

Dependent variable

Application-oriented specialty

Y

Whether to choose an applied major (1 = Yes; 0 = No)

0.462

0.499

Explanatory variables

Gender

Sex

Gender (1 = male; 0 = female)

0.493

0.500

Age

Age

Chronological age

19.199

0.850

Math scores

Math grades

Calculus grades

71.211

11.312

English scores

English grades

CET-4 scores

428.659

37.009

Total household income

Income

Annual household income level (1 = less than 150,000; 2 = 15-450,000; 3 = 450,000 or more)

1.876

0.653

Account type

Rural

Hukou type (1 = rural hukou; 0 = Urban hukou)

0.577

0.494

Hometown

Area

Area where the hometown is located (1 = West; 2 = Central; 3 = East)

2.277

0.760

2.4. Procedure

Data collection was conducted from January 2022 to March 2023. In the pre-survey stage, we mainly based on 117 respondents at Anhui University of Finance and Economics (respondents from freshman to senior year, covering all majors in the university, the actual survey of 120 people, 3 people were eliminated because they did not complete all the interviews), and with the permission of the interviewees, we conducted face-to-face interviews with the interviewees, and the main content of the interviews was to clarify the rationale, feasibility and use of the data collected later. Based on a field survey of 117 participants, we took into account various factors such as student grade level, school differences, household registration and geographical differences in student origin. In response to the feedback from this pre-survey, we optimized and adjusted the questions in our questionnaire to avoid problems in the pre-survey, and we improved the questionnaire method by improving the way we asked questions and the design of options. These changes are made to make it easier for attendees to complete questions efficiently and accurately.

Upon gathering a total of 1200 responses, the survey was concluded, ultimately obtaining 1164 valid questionnaires (the recovery rate of valid questionnaires reached 97%, 36 of the questionnaires were missing questions and were not complete, so these 36 questionnaires were excluded). Average completion time for the entire questionnaire was approximately 6 minutes.

The questionnaire consisted of two main parts. The first part included demographic information about the respondents, such as age, gender, grade level, hometown, and university. The second part contained information on major choice, family background, and other potential factors that could influence major choice. The questionnaire can be obtained by contacting the corresponding author.

2.5. Statistical Analyses

Linear Probability Model (LPM) and logit model can be used to predict the probability value P(y|x) of an event. However, scholars have engaged in heated discussions on which model is superior [40,41]. Linear probability models are based on the assumption of a linear relationship and are estimated using linear regression methods [42]. Logit model is a statistical method used to build binary classification models. It models the relationship between the independent and dependent variables as a logistic function that is used to predict the probability of the dependent variable [42]. In logistic regression, the dependent variable is usually represented as a binomial distribution representing two discrete possible outcomes, and the independent variable is either continuous or discrete. Some scholars argue that LPM is superior to logit because OLS regression is faster than logistic regression, probability changes are more intuitive than changes in odds ratios (P(y|x)/1-P(y|x)), and the meaning of parameters is more intuitive when interpreting the model [43]. Moreover, due to the problem of quasi-complete separation in the Maximum Likelihood Estimation (MLE) used by the logit model, the logit model may collapse, but the LPM model does not face this issue [44]. However, scholars who support the logit model believe that LPM may produce invalid probability prediction values, while logit does not, and the logit model is less affected by the interaction between explanatory variables, making the parameters more stable. The Linear Discriminant Model (LDM) designed by Allison [45] combines the advantages of LPM and logit. The LDM model is derived from the conversion of LPM and logit models. Firstly, LDM can be converted into a logit model, and secondly, the parameter estimation of LPM can be converted into the maximum likelihood estimation of LDM parameters. Therefore, effective prediction probabilities can be obtained by inserting the converted parameters into the logit model.

Suppose there exists a training set of n independent observations (xi , yi), i=1,2,...,n, such that for any pair (x , y) with given Y = j (j = 1, 2, ..., k), the distribution of X follows X ~ N (mj , σ). Let nj be the number of observed values in the cluster Y= j. In the binary case, where k = 2, we estimate the slope b and intercept a in the Linear Probability Model (LPM) using OLS, as described by Allison (2020) (Eq. 1).

      (1)

By following the parameter transformation derived by Haggstorm (1983), we can obtain the estimates α and β for the Logit model.

        (2)

In the specific data regression analysis, we applied the stata17.0 regression software for regression analysis.”

  1. National Bureau of Statistics of China, “China Education Yearbook,” People's Education Press, 2023.
  2. Y. Tang, M. Pu and H. Chen, “T. Beecher and Trollell: Academic tribes and their territories: knowledge exploration and disciplinary culture (retranslation),” Peking University Press, 2015, pp. 40-41.
  3. D. Mortens and C. Pissarides, “Unemployment responses to 'skill-biased' technology shocks: the role of labour market policy,” Economic Journal, 2005, vol. 109(455), pp. 242-265. https://doi.org/1111.1468/0297-00431.
  4. Y. Ding, L. Du, W. Li, Y. Wu, J. Yang and X. Ye, “The Impact of Information Intervention on College Entrance Examination Major Choices: Evidence from a Large Scale Randomized Experiment,” Quarterly Journal of Economics, 2021, vol. 21(6), pp. 2239-2262.
  5. S. Caudill, “PRACTITIONERS CORNER: An Advantage of the Linear Probability Model over Probit or Logit,” Oxford Bulletin of Economics & Statistics, 2010, vol. 50(4). https://doi.org/10.1111/j.1468-0084.1988.mp50004005.x
  6. C. Suneel and S. Galit, “Linear Probability Models (LPM) and Big Data: The Good, the Bad, and the Ugly,” SSRN Electronic Journal, 2013. http://dx.doi.org/2139.2353841/ssrn.
  7. J. James and B. Miriam, “Monte Carlo simulations using extant data to mimic populations: Applications to the modified linear probability model and logistic regression,” Psychological methods, 2021, vol. 26(4), pp. 450-465. https://doi.org/10.1037/met0000383.supp
  8. C. Joan-Timoneda, “Estimating group fixed effects in panel data with a binary dependent variable: How the LPM outperforms logistic regression in rare events data,” Social Science Research, 2021, vol. 93. https://doi.org/10.1016/j.ssresearch.2020.102486
  9. P. von Hippel and J. Workman, “From Kindergarten Through Second Grade, U.S. Children’s Obesity Prevalence Grows Only During Summer Vacations,” Obesity, 2016, vol. 24(11), pp. 2296–2300. https://doi.org/10.1002/oby.21613
  10. P Allison, “Better Predicted Probabilities from Linear Probability Models,” Statistical Horizons, 2020. https://statisticalhorizons.com/better-predicted-probabilities/
  11. Page 3-Page 5: Method, General: Research design section is completely missing and must be added.

Response 21: Thank you for bringing to our attention that the Research Design section is missing from our manuscript's Method section. We appreciate your careful review and identification of this oversight. We apologize for the omission and will promptly rectify it by adding a dedicated Research Design subsection to the Method section. This subsection will provide a comprehensive explanation of the design employed in our study, including its purpose and specific components. Thank you for highlighting this issue, as it will contribute to the overall improvement and completeness of our manuscript.

In lines 136 to 144 of the revised manuscript, the main changes are as follows:

“2.1. Research design

This study employed a quantitative research design with the aim of exploring gender differences in major choice. A large-scale representative sample was utilized, and data was collected through structured questionnaires obtained via social media platforms. The questionnaires included items related to demographic information, major aspirations, factors influencing major choice, and perceptions of gender-related barriers. Data analysis primarily utilized LDM to summarize and examine gender differences in major preferences, motivations, and cognitive barriers. The study adhered to ethical guidelines and ensured participant confidentiality and anonymity.”

  1. Page 3, Line 108-109: The citation/citations needed for following sentence “There are over 50   finance and economics universities in China, with a total of over 1 million undergraduate students.’’

Response 22: We apologize for not providing the necessary citations in the manuscript. To correct this, we provide a reliable source to verify the information provided. We appreciate your diligence in reviewing our manuscript and ensuring the accuracy of our citations.

In lines 147 to 148 of the revised manuscript, the main changes are as follows:

“There are over 50 finance and economics universities in China, with a total of over 1 million undergraduate students [7].”

  1. National Bureau of Statistics of China, “China Education Yearbook,” People's Education Press, 2023.
  2. Page 3, Line 117-119: In the following sentence ‘‘In our sample, the number of female students (n = 590) was slightly higher than that 117 of male students (n = 574). In terms of age, there was little difference between males and 118 females, with an average age of 19 years old (ranging from freshman to senior year, 17-27 years old).’’ Small ns representing subsample size must be italic. Authors must report age with two decimals with its standard deviation. Grade level distribution must be in a separate sentence.

Response 23: Thanks for the suggestion. We set the value representing the size of the specimen to italics, and we left the decimal of age to two decimal places, and added the standard deviation of age. Furthermore, we set the grade distribution of the sample in separate sentences.

In lines 195 to 198 of the revised manuscript, the main changes are as follows:

“In our sample, the number of female students (n = 590) was slightly higher than that of male students (n = 574). In terms of age, there was little difference between males and females, with an average age of 19.20 years old (The standard deviation is 0.85). Students ranged from freshman to senior, and respondents ranged in age from 17 to 27 years old.”

  1. Page 3, Line 122-123: In the following sentence “In terms of academic performance, female students performed better than male students, with an average score of 74 in Calculus, which was higher than the male students' average score of 69.” Authors must report average scores with two decimals e.g., 69.12.

Response 24: Thank you for pointing out the need to report the average score of two decimal places in our manuscript. To solve this problem, we modified the sentence to report the average score of two decimal places. We appreciate your attention to detail and will ensure that the results are accurately reported in our revised manuscript.

In lines 200 to 203 of the revised manuscript, the main changes are as follows:

“In terms of academic performance, female students performed better than male students, with an average score of 74.00 in Calculus, which was higher than the male students' average score of 69.00.”

  1. Page 3, Line 126-129: I think authors forget to add percentages following sentences. ‘‘Only 154 female students (add %) were from high-income families, while 131 male students (add %) were from high-income families. On the other hand, 222 female students (add %) were from low-income families, while only 107 male students (add %) were from low-income families.’’

Response 25: Thank you for bringing to our attention the omission of percentages in the sentences. We sincerely appreciate your careful review and feedback. We have supplemented the corresponding percentage. We apologize for the oversight and appreciate your valuable contribution toward improving the accuracy and clarity of our manuscript.

In lines 204 to 208 of the revised manuscript, the main changes are as follows:

“Only 54 female students (9.15%) were from high-income families, while 131 male students (22.82%) were from high-income families. On the other hand, 222 female students (37.63%) were from low-income families, while only 107 male students (18.64%) were from low-income families.”

  1. Page 3-Page 5: Method, General: Authors must give information about sampling procedure in Participants section (convenience sampling?)

Response 26: Thank you for your feedback regarding the need to provide information about the sampling procedure in the Participants section of our manuscript. We appreciate your attention to detail. In response to your suggestion, we will include a subsection within the Participants section to outline the sampling procedure used in our study. This will allow readers to better understand the method of participant recruitment and selection, including any specifics about convenience sampling or other relevant details. We acknowledge the importance of transparency in research methodology and appreciate your input in ensuring the completeness of our manuscript.

In lines 145 to 163 of the revised manuscript, the main changes are as follows:

“2.2. Participants

The participants in this study are undergraduate students studying at universities of finance and economics. There are over 50 finance and economics universities in China, with a total of over 1 million undergraduate students [7]. To ensure a comprehensive representation of universities with varying academic reputations and geo-graphical locations throughout the country, we extensively invited students from finance and economics universities situated in the Eastern, Central, and Western regions of China to participate in an online survey. The survey was distributed to a total of 1,200 students. Participating universities included Shandong University of Finance and Economics, Anhui University of Finance and Economics and Jiangxi University of Finance and Economics, among others.

The research data was collected via an online survey, ensuring the anonymity of the student participants. The online survey platform Wenjuanxing (https://www.wjx.cn) was utilized for data collection. The survey was formulated and disseminated via hyperlinks on popular Chinese social media platforms, namely Weibo, WeChat, and QQ. The dissemination of these links occurred with the assistance of teachers and classmates who subsequently forwarded them. While our survey methodology was not limited to traditional sampling techniques, we made efforts to maximize participant recruitment through diverse distribution channels to ensure sample diversity and representativeness.”

  1. National Bureau of Statistics of China, “China Education Yearbook,” People's Education Press, 2023.
  2. Page 3, Page 4 Line 133-142: Well done and written.

Response 27: We are delighted to hear that you found those sections well done and well-written. Positive feedback like yours serves as a great encouragement for us, and we appreciate your recognition.

  1. Page 4 Line 152: What is the reason for removing 36 participants add a reason sentence?

Response 28: We apologize for the lack of clarity in the manuscript. The decision to remove these 36 participants was based on their incomplete or inconsistent responses during data collection. By removing these participants, we aimed to ensure the integrity and reliability of the data analysis and results. We appreciate your attention to detail and your question, as it allows us to provide further clarification on this important aspect of our study.

In lines 227 to 230 of the revised manuscript, the main changes are as follows:

“Upon gathering a total of 1200 responses, the survey was concluded, ultimately obtaining 1164 valid questionnaires (the recovery rate of valid questionnaires reached 97%, 36 of the questionnaires were missing questions and were not complete, so these 36 questionnaires were excluded).”

  1. Page 4 Line 152:  Following must be at the end of Data collection process or Procedure section. ‘‘Average completion time for the entire questionnaire was approximately 6 minutes.’’

Response 29: We appreciate your input in improving the clarity and completeness of our manuscript. We will make the necessary adjustments and include the following sentence as suggested: “Average completion time for the entire questionnaire was approximately 6 minutes”. This will provide readers with important information about the time required for participants to complete the questionnaire. Thank you for bringing this to our attention and helping us enhance our manuscript.

In lines 230 to 231 of the revised manuscript, the main changes are as follows:

“Average completion time for the entire questionnaire was approximately 6 minutes.”

  1. Page 4 Line 166:   It is unclear what authors want to mean with micro-level scenarios?

Response 30: Thanks for the suggestion. We replaced the word “micro-level scenarios” with “specific scenarios” in the sentence.

In lines 169 to 170 of the revised manuscript, the main changes are as follows:

“Based on the definition of Beecher et al. [37], we consider majors that are vocational, technical, and involve specific scenarios as applied majors.”

  1. Y. Tang, M. Pu and H. Chen, “T. Beecher and Trollell: Academic tribes and their territories: knowledge exploration and disciplinary culture (retranslation),” Peking University Press, 2015, pp. 40-41.
  2. Page 4 Line 194:   Following information is completely wrong and must remove “When the dependent variable y is a binary or multinomial discrete variable in a linear regression model’’ true maybe generalized linear model wo of them is completely different things.

Response 31: We apologize for the mistake. To rectify the issue, we will remove the statement “When the dependent variable y is a binary or multinomial discrete variable in a linear regression model”, and revise it to accurately reflect that the analysis may require a generalized linear model instead.

In lines 238 to 239 of the revised manuscript, the main changes are as follows:

“Linear Probability Model (LPM) and logit model can be used to predict the probability value P(y|x) of an event.”

  1. Page 3-Page 5: Method, General: Authors must add which statistical software used in their analyses.

Response 32: Thanks for the suggestion. We apologize for the oversight in not explicitly mentioning the software utilized for the statistical analyses. In light of this feedback, we will update the Methods section to include information about the specific statistical software employed in our study. By including this information, readers will have a clear understanding of the software used to conduct the statistical analyses in our study. We appreciate your valuable input and will make the necessary revisions accordingly.

In lines 270 to 271 of the revised manuscript, the main changes are as follows:

“In the specific data regression analysis, we applied the Stata17.0 regression software for regression analysis.”

  1. Page 3-Page 5: Method, General: No need to Line 212-219 and must remove from manuscript.

Response 33: Thanks for your suggestion. We appreciate your input and understand your suggestion to remove that particular portion from the manuscript. Upon careful consideration, we have decided to remove lines 212-219 as per your recommendation. By removing these lines, we aim to streamline the content and improve the overall clarity and focus of the Method section. Thank you for bringing this to our attention, and we will ensure that the removal of lines 212-219 is reflected in the revised version of the manuscript.

  1. Page 6, Page 12, Results section general: Although Sustainability journal instructions did not give authors a systematic reporting for statistical results. In such cases, using a common format is a good choice such as APA 7. Moreover, along the manuscript reporting of statistics need to be completely correct according to APA 7 reporting.

Response 34: We appreciate your suggestions. Based on your suggestions, we have adopted the APA 7 format throughout the manuscript and carefully checked the relevant reporting statistics, this approach will ensure the accuracy and consistency of statistical reporting in line with APA guidelines. We have adjusted the values in Table 2.

In line 317 of the revised manuscript, the main changes are as follows:

“Table 2. Baseline regression results”

(1)

(2)

(3)

(4)

Sex

-2.769***

-2.744***

-2.680***

-2.624***

(0.151)

(0.159)

(0.160)

(0.167)

Age

-0.079

-0.089

(0.091)

(0.092)

Math grades

0.059***

0.060***

(0.008)

(0.008)

English grades

0.003

0.004

(0.002)

(0.002)

Income (medium level vs. low level)

-1.109***

-1.091***

(0.178)

(0.183)

Income (high level vs. low level)

-1.184***

-1.327***

(0.250)

(0.267)

Account

0.193

0.247

(0.154)

(0.162)

Area (East vs. West)

-0.911***

-0.921***

(0.220)

(0.228)

Area (Central vs. West)

-0.399*

-0.537**

(0.218)

(0.227)

Constant

1.041***

-3.020

2.257***

-1.807

(0.122)

(1.927)

(0.282)

(1.970)

LR chi2

435.891

518.911

502.283

582.474

Pseudo R2

0.271

0.323

0.313

0.363

Observations

1,164

1,164

1,164

1,164

  1. Page 6, Page 12, Results section general: Authors must report all findings with the same decimal. Apa recommend two but for more precision three decimal may be reported. Along this section authors must correct reporting of findings with two or three decimal. This comments is valid for tall tables in results section.

Response 35: We understand the importance of maintaining consistency and precision in reporting decimal places for all findings. In light of your suggestion, we will ensure that all findings reported in the Results section, including those presented in tables, are consistently reported with either two or three decimal places, depending on the level of precision required. By adhering to this approach, we aim to enhance clarity and accuracy in the presentation of our results.

In line 317 of the revised manuscript, the main changes are as follows:

“Table 2. Baseline regression results”

(1)

(2)

(3)

(4)

Sex

-2.769***

-2.744***

-2.680***

-2.624***

(0.151)

(0.159)

(0.160)

(0.167)

Age

-0.079

-0.089

(0.091)

(0.092)

Math grades

0.059***

0.060***

(0.008)

(0.008)

English grades

0.003

0.004

(0.002)

(0.002)

Income (medium level vs. low level)

-1.109***

-1.091***

(0.178)

(0.183)

Income (high level vs. low level)

-1.184***

-1.327***

(0.250)

(0.267)

Account

0.193

0.247

(0.154)

(0.162)

Area (East vs. West)

-0.911***

-0.921***

(0.220)

(0.228)

Area (Central vs. West)

-0.399*

-0.537**

(0.218)

(0.227)

Constant

1.041***

-3.020

2.257***

-1.807

(0.122)

(1.927)

(0.282)

(1.970)

LR chi2

435.891

518.911

502.283

582.474

Pseudo R2

0.271

0.323

0.313

0.363

Observations

1,164

1,164

1,164

1,164

  1. Page 6, Page 12, Results section general: Under the tables, instead of Authors must use Note.This comment is valid for tall tables in results section.

Response 36: Thank you for the suggestion. Thank you for bringing this to our attention, and we will make the necessary revisions to use the label "Note" under the tables in the Results section, specifically for tall tables.

In lines 318 to 320 of the revised manuscript, the main changes are as follows:

“Note. The numbers in parentheses are robust standard errors; *, **, *** indicate statistically significant differences at the 10%, 5%, and 1% levels, respectively. Column (1)-(4) show the regression results when different control variables are added.”

  1. Page 6, Page 12, Results section general: Statistical sybols writings must be corrected along the manuscript.

Response 37: We understand the importance of maintaining consistent and accurate presentation of statistical symbols in scientific writing. We will thoroughly review the manuscript and make the necessary corrections to ensure that all statistical symbols are properly formatted and in accordance with the appropriate conventions. Your feedback is valuable to us, and we appreciate your attention to detail. We will address this issue and ensure that the statistical symbols are correctly written throughout the manuscript.

In lines 317 to 320 of the revised manuscript, the main changes are as follows:

“Table 2. Baseline regression results

(1)

(2)

(3)

(4)

Sex

-2.769***

-2.744***

-2.680***

-2.624***

(0.151)

(0.159)

(0.160)

(0.167)

Age

-0.079

-0.089

(0.091)

(0.092)

Math grades

0.059***

0.060***

(0.008)

(0.008)

English grades

0.003

0.004

(0.002)

(0.002)

Income (medium level vs. low level)

-1.109***

-1.091***

(0.178)

(0.183)

Income (high level vs. low level)

-1.184***

-1.327***

(0.250)

(0.267)

Account

0.193

0.247

(0.154)

(0.162)

Area (East vs. West)

-0.911***

-0.921***

(0.220)

(0.228)

Area (Central vs. West)

-0.399*

-0.537**

(0.218)

(0.227)

Constant

1.041***

-3.020

2.257***

-1.807

(0.122)

(1.927)

(0.282)

(1.970)

LR chi2

435.891

518.911

502.283

582.474

Pseudo R2

0.271

0.323

0.313

0.363

Observations

1,164

1,164

1,164

1,164

Note. The numbers in parentheses are robust standard errors; *, **, *** indicate statistically significant differences at the 10%, 5%, and 1% levels, respectively. Column (1)-(4) show the regression results when different control variables are added.”

  1. Page 6, Page 12, Results section general: Authors must double check if figure 2 title is correct or not?

Response 38: We appreciate your valuable suggestions for our manuscript. We carefully reviewed the title of Figure 2 and revised it. Once again, we sincerely thank you for your expertise and feedback in improving the quality of our manuscripts.

In lines 334 to 335 of the revised manuscript, the main changes are as follows:

“Figure 2. The impact of gender differences on professional choice in different family contexts.”

  1. Page 6-Page 9 Results General: Results section and Discussion must be separate.

Response 39: We appreciate your valuable suggestions for our manuscript, we have carefully considered and revised them. Once again, we sincerely thank you for your expertise and feedback in improving the quality of our manuscripts.

In lines 336 to 438 of the revised manuscript, the main changes are as follows:

“4. Discussion and Conclusion

4.1. Discussion

The analysis allowed the RQs to be answered. The first RQ that aims to ascertain the relationship between gender and students' major choices within the context of broad discipline enrollment system backgrounds. Findings indicate significant gender differences in undergraduate students' major choices. This result aligns with the findings of Shao et al. (2022), who used administrative records from a prestigious medical school in China to investigate gender disparities in medical students' specialty choices. Existing research suggests that the disparities in innate capabilities between males and females, arising from biological and genetic factors, such as females being disadvantaged in abstract thinking and scientific cognition, manifest in the selection of disciplines and majors as students enter higher education. Consequently, gender differences in discipline and major choice are exogenous, objective, and unavoidably reflect pre-existing natural endowments [47]. This viewpoint has been extensively corroborated in research examining the gender impact on students' decisions to pursue science, technology, engineering, and mathematics (STEM) disciplines [22]. This study expands upon the existing literature by discussing gender disparities in major choices within the domain of universities of finance and economics.

Regarding the second RQ, the analysis confirms that the probability of female students choosing applied majors is significantly higher than that of male students, approximately 2.77 times higher. This finding is surprising as it appears to contradict some previous studies that reported no advantage for female students in entering popular advantageous majors [48]. Some studies suggest that differences in academic performance may be an important factor contributing to gender disparities in major choices [49]. In this study, even after controlling for academic performance and other potentially influencing variables, the analysis results robustly sup-port the significant tendency of female students to choose popular applied majors over male students. Shao et al pointed out various obstacles hindering female students from choosing high-paying surgical specialties when analyzing the likelihood of female students selecting such specialties. These obstacles include excessive physical demands, discrimination in recruitment, unfriendly work environments for women, and difficulties in balancing family responsibilities [49]. From the perspective of maintaining social gender segregation, both males and females, influenced by external factors and driven by internalized different values and potential social discrimination, make distinct decisions in education and major choices, tending towards the professions society expects of them [50]. In the current study, as the majors in finance and economics universities mainly comprise non-STEM disciplines such as accounting and finance, these majors are often considered more suitable for female students by society [23]. Therefore, in our research, female students tend to choose popular applied majors.

The last RQ pertains to the moderating effect of family background on the relationship between gender differences and major choices. This study finds that family background significantly influences the gender disparities in major choices. Specifically, females from economically developed regions and high-income households have a greater probability of choosing applied majors. A possible explanation is that females from economically developed regions and high-income households have easier access to resources and opportunities related to applied majors, making them more likely to choose these fields. On the other hand, in regions with relatively lower levels of economic development and income, females may face more social, economic, and cultural constraints, leading them to choose different types of majors. Additionally, female students from rural areas are more inclined to choose popular applied majors [51]. This finding is similar to a study by Yang & Sun [52], which showed that students from rural areas are more likely to choose applied majors than students from urban areas with broader employment prospects.

4.2. Limitations

Our study is not without limitations. The first limitation is that we only examined students who have already made their major choices and could not fully capture the mobility of students across different majors. Future longitudinal research could be conducted to investigate the causal relationship between gender and major choices and to explore the specific mechanisms driving these gender differences. The second limitation is that our study only focused on majors within financial and economic universities, which may limit the generalizability of our findings. To address this limitation, future research could expand the scope of the study to include comprehensive universities in order to provide a broader based evidence. Finally, the time period of our study is during the COVID-19 period, and it is too long compared to the time frame of most other studies, data collection may be affected by the pandemic, and students' choice of major may also be due to changes in the format of the curriculum due to the pandemic, such as from offline to online, resulting in changes in students' attitudes towards certain majors. These can lead to less robust regression results. Future research can explore the impact of these factors on undergraduate majors in finance and economics universities without or as little impact from the pandemic.

4.3. Practical Implications

Our findings have significant educational implications for promoting gender equality and social justice in China's higher education system, particularly, under the new broad discipline enrollment system. Females are more likely to enter popular applied majors, which can provide them with greater opportunities to alleviate their disadvantaged position in the labor market. However, it is important to acknowledge that applied majors in certain fields, such as finance and accounting, also have limitations in their employment direction. Increased numbers of female graduates in applied majors also create a risk of female employees facing intensified competitive pressures in these industries. Therefore, as long as gender stereotypes persist in the job market, gender equity issues such as opportunities and treatment should be highlighted. Meanwhile, interventions should work towards place great emphasis on guiding students in their major choices, providing them with adequate professional introductions and employment information to reduce students' lack of understanding or blind choice when selecting majors.

4.4. Conclusion

The literature on the major choices in higher education in China has mostly focused on the traditional enrollment model, therefore, there is little research analyzing the impact of gender on major choices under the broad discipline enrollment system. Against the backdrop of the emerging and widely promoted broad discipline enrollment reform in China in recent years, our research has advanced the existing literature. Specifically, the purpose of this study was to analyze how gender affects students’ choice of applied majors versus non-applied majors. This includes the degree to which gender differences have an impact on professional choices, and whether this impact will vary depending on family background and region. In recent research, gender classifications have become increasingly diverse. Due to our inability to analyze gender diversity, we follow Cui et al. [53] in categorizing gender mainly as male and female. By using the LDM model to analyze who was more likely to enter advantaged applied majors in universities, we focus on gender differences in professional education resources during the undergraduate stage. Furthermore, we explored the differential impact of gender on major choices due to differences in family background (such as income level, urban and rural areas, and regions). Our study provided empirical evidence that, under the broad discipline enrollment system, the probability of female students choosing applied majors is significantly higher than that of male students and this gap increases with the increase of family income levels. Furthermore, female students from Eastern regions and rural areas in China are more likely to choose applied majors than females in other areas.”

  1. 22. D. Li, Y. Wang and L. Li, “Educational choice has greater effects on sex ratios of college STEM majors than has the greater male variance in general intelligence (g),” Intelligence,
  2. X. Shao and T. Wang, “Grey’s Anatomy: Gender Differences in Specialty Choice for Medical Students in China,” Sustainability, 2022, vol. 14(1). https://doi.org/10.3390/su14010230
  3. A. Lina and N. Emma, “Culture and the gender gap in choice of major: An analysis using sibling comparisons,” Journal of Economic Behavior & Organization, 2022, vol. 201, pp. 346-373. https://doi.org/10.1016/j.jebo.2022.07.026
  4. A. Kimberlee Shauman and H. Jill, “Gender, race-ethnicity and postdoctoral hiring in STEMM fields,” Social Science Research, 2023, vol. 113. https://doi.org/10.1016/j.ssresearch.2023.102854
  5. S. Stijn, W. Bart, F. Lot, R. Elisabeth, D. Eva, F. Filip-De and D. Wouter, “How interest fit relates to STEM study choice: Female students fit their choices better,” Journal of Vocational Behavior, 2021, vol. 129. https://doi.org/10.1016/j.jvb.2021.103614
  6. R. Gareth and S. Volker, “Gender-based segregation in education, jobs and earnings in South Africa,” World Development Perspectives, 2021, vol. 23. https://doi.org/10.1016/j.wdp.2021.100348
  7. Y. Zheng, C. Lin, C. Jason Chang and Y. Liao, “Who is able to choose? A meta-analysis and systematic review of the effects of family socioeconomic status on school choice,” International Journal of Educational Research, 2022, vol. 112. https://doi.org/10.1016/j.ijer.2022.101943
  8. S. Yang and M. Sun, “Family Background, Major Selection, and Income Return: An Empirical Study Based on the China General Social Survey (CGSS) Data,” Northwest Population Journal, 2022, vol. 41(02), pp. 52-66.
  9. N. Cui, R. Wang, F. Song and J. Jin, “Experiences and Perceptions of Male Nursing Students in a Single-Sex Class: A Qualitative Descriptive Study,” Nurse Education in Practice, 2021, vol. 51. https://doi.org/10.1016/j.nepr.2021.102996
  10. Page 6-Page 9 Results General: Authors must construct Discussion sectionwith following subtitles Discussion, Limitations, Practical implications, Conclusion and must move all related information to related sections.

Response 40: We appreciate your valuable suggestions for our manuscript, we have carefully considered and revised them. Once again, we sincerely thank you for your expertise and feedback in improving the quality of our manuscripts.

In lines 337 to 417 of the revised manuscript, the main changes are as follows:

4.1. Discussion

The analysis allowed the RQs to be answered. The first RQ that aims to ascertain the relationship between gender and students' major choices within the context of broad discipline enrollment system backgrounds. Findings indicate significant gender differences in undergraduate students' major choices. This result aligns with the findings of Shao et al. (2022), who used administrative records from a prestigious medical school in China to investigate gender disparities in medical students' specialty choices. Existing research suggests that the disparities in innate capabilities between males and females, arising from biological and genetic factors, such as females being disadvantaged in abstract thinking and scientific cognition, manifest in the selection of disciplines and majors as students enter higher education. Consequently, gender differences in discipline and major choice are exogenous, objective, and unavoidably reflect pre-existing natural endowments [47]. This viewpoint has been extensively corroborated in research examining the gender impact on students' decisions to pursue science, technology, engineering, and mathematics (STEM) disciplines [22]. This study expands upon the existing literature by discussing gender disparities in major choices within the domain of universities of finance and economics.

Regarding the second RQ, the analysis confirms that the probability of female students choosing applied majors is significantly higher than that of male students, approximately 2.77 times higher. This finding is surprising as it appears to contradict some previous studies that reported no advantage for female students in entering popular advantageous majors [48]. Some studies suggest that differences in academic performance may be an important factor contributing to gender disparities in major choices [49]. In this study, even after controlling for academic performance and other potentially influencing variables, the analysis results robustly sup-port the significant tendency of female students to choose popular applied majors over male students. Shao et al pointed out various obstacles hindering female students from choosing high-paying surgical specialties when analyzing the likelihood of female students selecting such specialties. These obstacles include excessive physical demands, discrimination in recruitment, unfriendly work environments for women, and difficulties in balancing family responsibilities [49]. From the perspective of maintaining social gender segregation, both males and females, influenced by external factors and driven by internalized different values and potential social discrimination, make distinct decisions in education and major choices, tending towards the professions society expects of them [50]. In the current study, as the majors in finance and economics universities mainly comprise non-STEM disciplines such as accounting and finance, these majors are often considered more suitable for female students by society [23]. Therefore, in our research, female students tend to choose popular applied majors.

The last RQ pertains to the moderating effect of family background on the relationship between gender differences and major choices. This study finds that family background significantly influences the gender disparities in major choices. Specifically, females from economically developed regions and high-income households have a greater probability of choosing applied majors. A possible explanation is that females from economically developed regions and high-income households have easier access to resources and opportunities related to applied majors, making them more likely to choose these fields. On the other hand, in regions with relatively lower levels of economic development and income, females may face more social, economic, and cultural constraints, leading them to choose different types of majors. Additionally, female students from rural areas are more inclined to choose popular applied majors [51]. This finding is similar to a study by Yang & Sun [52], which showed that students from rural areas are more likely to choose applied majors than students from urban areas with broader employment prospects.

4.2. Limitations

Our study is not without limitations. The first limitation is that we only examined students who have already made their major choices and could not fully capture the mobility of students across different majors. Future longitudinal research could be conducted to investigate the causal relationship between gender and major choices and to explore the specific mechanisms driving these gender differences. The second limitation is that our study only focused on majors within financial and economic universities, which may limit the generalizability of our findings. To address this limitation, future research could expand the scope of the study to include comprehensive universities in order to provide a broader based evidence. Finally, the time period of our study is during the COVID-19 period, and it is too long compared to the time frame of most other studies, data collection may be affected by the pandemic, and students' choice of major may also be due to changes in the format of the curriculum due to the pandemic, such as from offline to online, resulting in changes in students' attitudes towards certain majors. These can lead to less robust regression results. Future research can explore the impact of these factors on undergraduate majors in finance and economics universities without or as little impact from the pandemic.

4.3. Practical Implications

Our findings have significant educational implications for promoting gender equality and social justice in China's higher education system, particularly, under the new broad discipline enrollment system. Females are more likely to enter popular applied majors, which can provide them with greater opportunities to alleviate their disadvantaged position in the labor market. However, it is important to acknowledge that applied majors in certain fields, such as finance and accounting, also have limitations in their employment direction. Increased numbers of female graduates in applied majors also create a risk of female employees facing intensified competitive pressures in these industries. Therefore, as long as gender stereotypes persist in the job market, gender equity issues such as opportunities and treatment should be highlighted. Meanwhile, interventions should work towards place great emphasis on guiding students in their major choices, providing them with adequate professional introductions and employment information to reduce students' lack of understanding or blind choice when selecting majors.”

  1. 22. D. Li, Y. Wang and L. Li, “Educational choice has greater effects on sex ratios of college STEM majors than has the greater male variance in general intelligence (g),” Intelligence,
  2. X. Shao and T. Wang, “Grey’s Anatomy: Gender Differences in Specialty Choice for Medical Students in China,” Sustainability, 2022, vol. 14(1). https://doi.org/10.3390/su14010230
  3. A. Lina and N. Emma, “Culture and the gender gap in choice of major: An analysis using sibling comparisons,” Journal of Economic Behavior & Organization, 2022, vol. 201, pp. 346-373. https://doi.org/10.1016/j.jebo.2022.07.026
  4. A. Kimberlee Shauman and H. Jill, “Gender, race-ethnicity and postdoctoral hiring in STEMM fields,” Social Science Research, 2023, vol. 113. https://doi.org/10.1016/j.ssresearch.2023.102854
  5. S. Stijn, W. Bart, F. Lot, R. Elisabeth, D. Eva, F. Filip-De and D. Wouter, “How interest fit relates to STEM study choice: Female students fit their choices better,” Journal of Vocational Behavior, 2021, vol. 129. https://doi.org/10.1016/j.jvb.2021.103614
  6. R. Gareth and S. Volker, “Gender-based segregation in education, jobs and earnings in South Africa,” World Development Perspectives, 2021, vol. 23. https://doi.org/10.1016/j.wdp.2021.100348
  7. Y. Zheng, C. Lin, C. Jason Chang and Y. Liao, “Who is able to choose? A meta-analysis and systematic review of the effects of family socioeconomic status on school choice,” International Journal of Educational Research, 2022, vol. 112. https://doi.org/10.1016/j.ijer.2022.101943
  8. S. Yang and M. Sun, “Family Background, Major Selection, and Income Return: An Empirical Study Based on the China General Social Survey (CGSS) Data,” Northwest Population Journal, 2022, vol. 41(02), pp. 52-66.

Reviewer 2 Report

Dear authors,

I was glad to review your paper, which is interesting and stimulates a reader’s mind. There are some strong elements, such as the presentation of the social-cultural context and the findings, which are easy to understand. However, there are some points that should be taken into consideration. To begin with, some more details could we be provided with regarding the methodology used and the sampling process followed at the abstract part. In addition some more information related to the methodology instrument, the reliability and the ethics are necessary as well at the methodology part. Please, it is also important for me to see some more in-text citations at the methodology part. At the discussion part some comparisons of your findings with those of some other similar ones should be made. Ceteris paribus should be also more explained.  Finally, there are some minor mistakes in the way the reference list is written.

Author Response

Response to comments by Reviewer #2

Dear authors,

I was glad to review your paper, which is interesting and stimulates a reader’s mind. There are some strong elements, such as the presentation of the social-cultural context and the findings, which are easy to understand. However, there are some points that should be taken into consideration.

Response: Thanks for recognizing and kindly considering this manuscript. Our response is as follows. The original comment is printed in black, followed by a reply in blue.

To begin with, some more details could we be provided with regarding the methodology used and the sampling process followed at the abstract part.

Response: Thank you for your valuable suggestions for the methodology and abstract parts of our manuscript. We have carefully considered your proposal and made the necessary modifications. Once again, we sincerely appreciate your expertise and feedback in improving the quality of our manuscript.

In lines 12 to 16 of the revised manuscript, the main changes are as follows:

“Based on a web survey of undergraduate students in China's broad enrollment context, this study uses Wenjuanxing (https://www.wjx.cn) to collect information by posting questionnaires on social media platforms, and analyzes the impact of gender differences on the major choices of finance and economics undergraduates by using the Linear Discriminant Model (LDM).”

In lines 137 to 144 of the revised manuscript, the main changes are as follows:

“This study employed a quantitative research design with the aim of exploring gender differences in major choice. A large-scale representative sample was utilized, and data was collected through structured questionnaires obtained via social media platforms. The questionnaires included items related to demographic information, major aspirations, factors influencing major choice, and perceptions of gender-related barriers. Data analysis primarily utilized LDM to summarize and examine gender differences in major preferences, motivations, and cognitive barriers. The study adhered to ethical guidelines and ensured participant confidentiality and anonymity.”

In addition some more information related to the methodology instrument, the reliability and the ethics are necessary as well at the methodology part. Please, it is also important for me to see some more in-text citations at the methodology part.

Response: Thank you for your further valuable suggestions for the methodology section of our manuscript. We have carefully considered your suggestions and made the necessary changes to the content of this section. Once again, we sincerely thank you for your expertise and feedback in improving the quality of our manuscripts.

In lines 136 to 236 of the revised manuscript, the main changes are as follows:

“This study employed a quantitative research design with the aim of exploring gender differences in major choice. A large-scale representative sample was utilized, and data was collected through structured questionnaires obtained via social media platforms. The questionnaires included items related to demographic information, major aspirations, factors influencing major choice, and perceptions of gender-related barriers. Data analysis primarily utilized LDM to summarize and examine gender differences in major preferences, motivations, and cognitive barriers. The study adhered to ethical guidelines and ensured participant confidentiality and anonymity.

... There are over 50 finance and economics universities in China, with a total of over 1 million undergraduate students [7]....

...Based on the definition of Beecher et al. [37], we consider majors that are vocational, technical, and involve specific scenarios as applied majors. Applied disciplines are characterized by strong practicality and skill requirements, higher alignment with the market, better fitting of enterprise skill requirements, adaptability to complex work needs, and more lucrative returns in the labor market [38]....

Because other factors may affect the choice of undergraduate majors, according to Ding et al. [39], we chose as explanatory or independent variables the personal and family characteristics of undergraduate students....”

  1. National Bureau of Statistics of China, “China Education Yearbook,” People's Education Press, 2023.
  2. Y. Tang, M. Pu and H. Chen, “T. Beecher and Trollell: Academic tribes and their territories: knowledge exploration and disciplinary culture (retranslation),” Peking University Press, 2015, pp. 40-41.
  3. D. Mortens and C. Pissarides, “Unemployment responses to 'skill-biased' technology shocks: the role of labour market policy,” Economic Journal, 2005, vol. 109(455), pp. 242-265. https://doi.org/1111.1468/0297-00431.
  4. Y. Ding, L. Du, W. Li, Y. Wu, J. Yang and X. Ye, “The Impact of Information Intervention on College Entrance Examination Major Choices: Evidence from a Large Scale Randomized Experiment,” Quarterly Journal of Economics, 2021, vol. 21(6), pp. 2239-2262.

At the discussion part some comparisons of your findings with those of some other similar ones should be made.

Response: Thank you for your valuable suggestions on the discussion section of our manuscript. We have carefully considered your suggestions and made the necessary changes to this section. Once again, we sincerely thank you for your expertise and feedback in improving the quality of our manuscripts.

In lines 337 to 386 of the revised manuscript, the main changes are as follows:

“4.1. Discussion

The analysis allowed the RQs to be answered. The first RQ that aims to ascertain the relationship between gender and students' major choices within the context of broad discipline enrollment system backgrounds. Findings indicate significant gender differences in undergraduate students' major choices. This result aligns with the findings of Shao et al. (2022), who used administrative records from a prestigious medical school in China to investigate gender disparities in medical students' specialty choices. Existing research suggests that the disparities in innate capabilities between males and females, arising from biological and genetic factors, such as females being disadvantaged in abstract thinking and scientific cognition, manifest in the selection of disciplines and majors as students enter higher education. Consequently, gender differences in discipline and major choice are exogenous, objective, and unavoidably reflect pre-existing natural endowments [47]. This viewpoint has been extensively corroborated in research examining the gender impact on students' decisions to pursue science, technology, engineering, and mathematics (STEM) disciplines [22]. This study expands upon the existing literature by discussing gender disparities in major choices within the domain of universities of finance and economics.

Regarding the second RQ, the analysis confirms that the probability of female students choosing applied majors is significantly higher than that of male students, approximately 2.77 times higher. This finding is surprising as it appears to contradict some previous studies that reported no advantage for female students in entering popular advantageous majors [48]. Some studies suggest that differences in academic performance may be an important factor contributing to gender disparities in major choices [49]. In this study, even after controlling for academic performance and other potentially influencing variables, the analysis results robustly sup-port the significant tendency of female students to choose popular applied majors over male students. Shao et al pointed out various obstacles hindering female students from choosing high-paying surgical specialties when analyzing the likelihood of female students selecting such specialties. These obstacles include excessive physical demands, discrimination in recruitment, unfriendly work environments for women, and difficulties in balancing family responsibilities [49]. From the perspective of maintaining social gender segregation, both males and females, influenced by external factors and driven by internalized different values and potential social discrimination, make distinct decisions in education and major choices, tending towards the professions society expects of them [50]. In the current study, as the majors in finance and economics universities mainly comprise non-STEM disciplines such as accounting and finance, these majors are often considered more suitable for female students by society [23]. Therefore, in our research, female students tend to choose popular applied majors.

The last RQ pertains to the moderating effect of family background on the relationship between gender differences and major choices. This study finds that family background significantly influences the gender disparities in major choices. Specifically, females from economically developed regions and high-income households have a greater probability of choosing applied majors. A possible explanation is that females from economically developed regions and high-income households have easier access to resources and opportunities related to applied majors, making them more likely to choose these fields. On the other hand, in regions with relatively lower levels of economic development and income, females may face more social, economic, and cultural constraints, leading them to choose different types of majors. Additionally, female students from rural areas are more inclined to choose popular applied majors [51]. This finding is similar to a study by Yang & Sun [52], which showed that students from rural areas are more likely to choose applied majors than students from urban areas with broader employment prospects.”

  1. 22. D. Li, Y. Wang and L. Li, “Educational choice has greater effects on sex ratios of college STEM majors than has the greater male variance in general intelligence (g),” Intelligence,
  2. X. Shao and T. Wang, “Grey’s Anatomy: Gender Differences in Specialty Choice for Medical Students in China,” Sustainability, 2022, vol. 14(1). https://doi.org/10.3390/su14010230
  3. A. Lina and N. Emma, “Culture and the gender gap in choice of major: An analysis using sibling comparisons,” Journal of Economic Behavior & Organization, 2022, vol. 201, pp. 346-373. https://doi.org/10.1016/j.jebo.2022.07.026
  4. A. Kimberlee Shauman and H. Jill, “Gender, race-ethnicity and postdoctoral hiring in STEMM fields,” Social Science Research, 2023, vol. 113. https://doi.org/10.1016/j.ssresearch.2023.102854
  5. S. Stijn, W. Bart, F. Lot, R. Elisabeth, D. Eva, F. Filip-De and D. Wouter, “How interest fit relates to STEM study choice: Female students fit their choices better,” Journal of Vocational Behavior, 2021, vol. 129. https://doi.org/10.1016/j.jvb.2021.103614
  6. R. Gareth and S. Volker, “Gender-based segregation in education, jobs and earnings in South Africa,” World Development Perspectives, 2021, vol. 23. https://doi.org/10.1016/j.wdp.2021.100348
  7. Y. Zheng, C. Lin, C. Jason Chang and Y. Liao, “Who is able to choose? A meta-analysis and systematic review of the effects of family socioeconomic status on school choice,” International Journal of Educational Research, 2022, vol. 112. https://doi.org/10.1016/j.ijer.2022.101943
  8. S. Yang and M. Sun, “Family Background, Major Selection, and Income Return: An Empirical Study Based on the China General Social Survey (CGSS) Data,” Northwest Population Journal, 2022, vol. 41(02), pp. 52-66.

Ceteris paribus should be also more explained.

Response: Thank you for your valuable suggestions for this part of our manuscript. "Ceteris paribus" is a Latin phrase that means "all other things being equal" or "holding everything else constant." It is often used in economics, social sciences, and other fields to isolate the effect of a particular variable by assuming that all other relevant factors remain unchanged. This concept allows researchers to analyze the impact of a specific factor or variable on a given outcome. In addition, we have revised the research question to include "Ceteris paribus" here based on your suggestions and those of other reviewers. Once again, we sincerely thank you for your expertise and feedback in improving the quality of our manuscripts.

In lines 123 to 124 of the revised manuscript, the main changes are as follows:

“RQ2: To what extent does gender influence the profession relative to other factors influencing the choice of major?”

  Finally, there are some minor mistakes in the way the reference list is written.

Response: Thank you for your valuable suggestions for the references in our manuscript. We have carefully considered your suggestions and carefully checked and revised the way the references are written. Once again, we sincerely thank you for your expertise and feedback in improving the quality of our manuscripts.

Reviewer 3 Report

Dear author,

first of all, I would like to praise this article for its clarity, nuance, and structure. I do have some suggestions for further improvement. You can find them in the attached document and can be summarised as follows:

I would advise you to rephrase the title: analyze the influence of gender differences in analyzing the scope of gender differences or simply gender effects to avoid implying a causal explanation one can only get by a controlled experiment.

I would omit to mention international students in the Introduction section since they are not included (or mentioned) in the Participants section and the study is already narrowed down to a very narrow sample that shares cultural and contextual variables.

I advise you to hypothesize, in limitations, different variables you should've controlled for but haven't. Also, explain why you chose those 3 you mentioned in Aims.

Please operationalize Ceteris paribus in Goals.

The Participants section can be improved by adding return rate (send vs. received questionnaires), region ratio, etc. Also, some missing values are in the brackets you intend to add.

Please elaborate on the sampling for the pre-survey.

Before proceeding to correlation analysis, you should consider addressing assumptions for LDM.

The Discussion section is too short, and abrupt and lacks some substantial explanation of, what appears to be, contradictory findings. In section 3.2. you state students from families with lower income levels are more likely to choose applied majors but in section 3.3. it is the reverse for females jet somehow they are more likely to choose an applied major if they are from rural areas that are considered to be more low-income. 

Limitation and Implication section could be improved slightly. 

Author Response

Response to comments by Reviewer #3

Dear author,

first of all, I would like to praise this article for its clarity, nuance, and structure. I do have some suggestions for further improvement. You can find them in the attached document and can be summarised as follows:

Response: Thank you for your recognition of our manuscript. Thank you also for your thoughtful comments and constructive suggestions. This has greatly helped us improve our paper. Our response is as follows. The original comment is printed in black, followed by a reply in blue.

I would advise you to rephrase the title: analyze the influence of gender differences in analyzing the scope of gender differences or simply gender effects to avoid implying a causal explanation one can only get by a controlled experiment.

Response: Thank you for your valuable suggestions on the title of our manuscript. We carefully considered your suggestions and those of other reviewers and revised the title of our manuscript. Once again, we sincerely thank you for your expertise and feedback in improving the quality of our manuscripts.

In lines 2 to 4 of the revised manuscript, the main changes are as follows:

“An analysis of Factors Influencing Chinese University Students' Major Choice from the Perspective of Gender Differences”

I would omit to mention international students in the Introduction section since they are not included (or mentioned) in the Participants section and the study is already narrowed down to a very narrow sample that shares cultural and contextual variables.

Response: We appreciate your valuable suggestions for our manuscript, and we have carefully considered and revised the original text based on your suggestions. Once again, we sincerely thank you for your expertise and feedback in improving the quality of our manuscripts.

In lines 33 to 34 of the revised manuscript, the main changes are as follows:

“Under this opening up, China's higher education has experienced unprecedented rapid development [6].”

  1. J. Zhang and Y. Wang, “Achievements of China's Higher Education Since the Reform and Opening up and Their Enlightenment to ‘Double First-Class’ Construction Path,” Journal of Tianjin University (Social Sciences), 2021, pp. 50-57.

I advise you to hypothesize, in limitations, different variables you should've controlled for but haven't. Also, explain why you chose those 3 you mentioned in Aims.

Response: We appreciate your valuable suggestions for our manuscripts, and we consider and respond carefully to your suggestions. For the suggestion of a related variable that we should control but do not control, we describe it in the content and table in the measures section of the manuscript. As for why we mention these three questions in our objectives, these three questions are the focus of our research and the questions around which our entire manuscript revolves and needs to be explained. We draw on the research of our predecessors in academia and select these as control variables to better achieve our research goals. In addition, we have revised and adjusted these three questions based on suggestions from other reviewers. Once again, we sincerely thank you for your expertise and feedback in improving the quality of our manuscripts.

In lines 116 to 126 of the revised manuscript, the main changes are as follows:

“In order to advance existing research, this study aimed to investigate the relationship between gender differences and undergraduate major choice in a broad-based admissions process of finance and economics university students by empirically testing the influence of gender on students’ choice. Further, specifically this study sought to answer the following three research questions (RQ):

RQ1: Under the current broad discipline enrollment system in China, will gender differences affect major choices in finance and economics?

RQ2: To what extent does gender influence the profession relative to other factors influencing the choice of major?

RQ3: Will the impact of gender differences on professional choices vary depending on family background?”

Please operationalize Ceteris paribus in Goals.

Response: Thank you for your valuable suggestions for this part of our manuscript. “Ceteris paribus” is a Latin phrase that means “all other things being equal” or “holding everything else constant”. It is often used in economics, social sciences, and other fields to isolate the effect of a particular variable by assuming that all other relevant factors remain unchanged. This concept allows researchers to analyze the impact of a specific factor or variable on a given outcome. In addition, we have revised the research question to include “Ceteris paribus” here based on your suggestions and those of other reviewers. Once again, we sincerely thank you for your expertise and feedback in improving the quality of our manuscripts.

In lines 123 to 124 of the revised manuscript, the main changes are as follows:

“RQ2: To what extent does gender influence the profession relative to other factors influencing the choice of major?”

The Participants section can be improved by adding return rate (send vs. received questionnaires), region ratio, etc. Also, some missing values are in the brackets you intend to add.

Response: We appreciate your valuable suggestions for this part of us, we listened carefully to your suggestions, and based on your comments and other reviewers, we put the references in your suggestions in the procedure section. Once again, we sincerely thank you for your expertise and feedback in improving the quality of our manuscripts.

In lines 227 to 231 of the revised manuscript, the main changes are as follows:

“Upon gathering a total of 1200 responses, the survey was concluded, ultimately obtaining 1164 valid questionnaires(the recovery rate of valid questionnaires reached 97%, 36 of the questionnaires were missing questions and were not complete, so these 36 questionnaires were excluded). Average completion time for the entire questionnaire was approximately 6 minutes.”

Please elaborate on the sampling for the pre-survey.

Response: We appreciate your valuable suggestions on the pre-investigation part of our manuscript, and we carefully consider and explain the relevant information of the pre-investigation in detail based on your suggestions. Once again, we sincerely thank you for your expertise and feedback in improving the quality of our manuscripts.

In lines 213 to 226 of the revised manuscript, the main changes are as follows:

“Data collection was conducted from January 2022 to March 2023. In the pre-survey stage, we mainly based on 117 respondents at Anhui University of Finance and Economics (respondents from freshman to senior year, covering all majors in the university, the actual survey of 120 people, 3 people were eliminated because they did not complete all the interviews), and with the permission of the interviewees, we conducted face-to-face interviews with the interviewees, and the main content of the interviews was to clarify the rationale, feasibility and use of the data collected later. Based on a field survey of 117 participants, we took into account various factors such as student grade level, school differences, household registration and geographical differences in student origin. In response to the feedback from this pre-survey, we optimized and adjusted the questions in our questionnaire to avoid problems in the pre-survey, and we improved the questionnaire method by improving the way we asked questions and the design of options. These changes are made to make it easier for attendees to complete questions efficiently and accurately.”

Before proceeding to correlation analysis, you should consider addressing assumptions for LDM

Response: Thank you for your suggestion regarding addressing assumptions for the Linear Regression Model (LDM) before conducting correlation analysis. Addressing the assumptions of the LDM is indeed an important step in statistical analysis. It helps ensure the validity and reliability of the results. In order to proceed with the correlation analysis, it would be prudent to first assess and satisfy the assumptions of the LDM, which include the assumptions of linearity, independence, homoscedasticity, and normality of residuals. We appreciate your insight and recommendation, and we make sure to address and satisfy the assumptions of the LDM before conducting the correlation analysis.

The Discussion section is too short, and abrupt and lacks some substantial explanation of, what appears to be, contradictory findings. In section 3.2. you state students from families with lower income levels are more likely to choose applied majors but in section 3.3. it is the reverse for females jet somehow they are more likely to choose an applied major if they are from rural areas that are considered to be more low-income.

Response: We appreciate your valuable suggestions for the discussion section of our manuscript, which we have carefully considered and revised in the light of your suggestions and those of other reviewers. Once again, we sincerely thank you for your expertise and feedback in improving the quality of our manuscripts.

In lines 337 to 386 of the revised manuscript, the main changes are as follows:

“4.1. Discussion

The analysis allowed the RQs to be answered. The first RQ that aims to ascertain the relationship between gender and students' major choices within the context of broad discipline enrollment system backgrounds. Findings indicate significant gender differences in undergraduate students' major choices. This result aligns with the findings of Shao et al. (2022), who used administrative records from a prestigious medical school in China to investigate gender disparities in medical students' specialty choices. Existing research suggests that the disparities in innate capabilities between males and females, arising from biological and genetic factors, such as females being disadvantaged in abstract thinking and scientific cognition, manifest in the selection of disciplines and majors as students enter higher education. Consequently, gender differences in discipline and major choice are exogenous, objective, and unavoidably reflect pre-existing natural endowments [47]. This viewpoint has been extensively corroborated in research examining the gender impact on students' decisions to pursue science, technology, engineering, and mathematics (STEM) disciplines [22]. This study expands upon the existing literature by discussing gender disparities in major choices within the domain of universities of finance and economics.

Regarding the second RQ, the analysis confirms that the probability of female students choosing applied majors is significantly higher than that of male students, approximately 2.77 times higher. This finding is surprising as it appears to contradict some previous studies that reported no advantage for female students in entering popular advantageous majors [48]. Some studies suggest that differences in academic performance may be an important factor contributing to gender disparities in major choices [49]. In this study, even after controlling for academic performance and other potentially influencing variables, the analysis results robustly sup-port the significant tendency of female students to choose popular applied majors over male students. Shao et al pointed out various obstacles hindering female students from choosing high-paying surgical specialties when analyzing the likelihood of female students selecting such specialties. These obstacles include excessive physical demands, discrimination in recruitment, unfriendly work environments for women, and difficulties in balancing family responsibilities [49]. From the perspective of maintaining social gender segregation, both males and females, influenced by external factors and driven by internalized different values and potential social discrimination, make distinct decisions in education and major choices, tending towards the professions society expects of them [50]. In the current study, as the majors in finance and economics universities mainly comprise non-STEM disciplines such as accounting and finance, these majors are often considered more suitable for female students by society [23]. Therefore, in our research, female students tend to choose popular applied majors.

The last RQ pertains to the moderating effect of family background on the relationship between gender differences and major choices. This study finds that family background significantly influences the gender disparities in major choices. Specifically, females from economically developed regions and high-income households have a greater probability of choosing applied majors. A possible explanation is that females from economically developed regions and high-income households have easier access to resources and opportunities related to applied majors, making them more likely to choose these fields. On the other hand, in regions with relatively lower levels of economic development and income, females may face more social, economic, and cultural constraints, leading them to choose different types of majors. Additionally, female students from rural areas are more inclined to choose popular applied majors [51]. This finding is similar to a study by Yang & Sun [52], which showed that students from rural areas are more likely to choose applied majors than students from urban areas with broader employment prospects.”

  1. 22. D. Li, Y. Wang and L. Li, “Educational choice has greater effects on sex ratios of college STEM majors than has the greater male variance in general intelligence (g),” Intelligence,
  2. X. Shao and T. Wang, “Grey’s Anatomy: Gender Differences in Specialty Choice for Medical Students in China,” Sustainability, 2022, vol. 14(1). https://doi.org/10.3390/su14010230
  3. A. Lina and N. Emma, “Culture and the gender gap in choice of major: An analysis using sibling comparisons,” Journal of Economic Behavior & Organization, 2022, vol. 201, pp. 346-373. https://doi.org/10.1016/j.jebo.2022.07.026
  4. A. Kimberlee Shauman and H. Jill, “Gender, race-ethnicity and postdoctoral hiring in STEMM fields,” Social Science Research, 2023, vol. 113. https://doi.org/10.1016/j.ssresearch.2023.102854
  5. S. Stijn, W. Bart, F. Lot, R. Elisabeth, D. Eva, F. Filip-De and D. Wouter, “How interest fit relates to STEM study choice: Female students fit their choices better,” Journal of Vocational Behavior, 2021, vol. 129. https://doi.org/10.1016/j.jvb.2021.103614
  6. R. Gareth and S. Volker, “Gender-based segregation in education, jobs and earnings in South Africa,” World Development Perspectives, 2021, vol. 23. https://doi.org/10.1016/j.wdp.2021.100348
  7. Y. Zheng, C. Lin, C. Jason Chang and Y. Liao, “Who is able to choose? A meta-analysis and systematic review of the effects of family socioeconomic status on school choice,” International Journal of Educational Research, 2022,vol. 112. https://doi.org/10.1016/j.ijer.2022.101943
  8. S. Yang and M. Sun, “Family Background, Major Selection, and Income Return: An Empirical Study Based on the China General Social Survey (CGSS) Data,” Northwest Population Journal, 2022, vol. 41(02), pp. 52-66.

Limitation and Implication section could be improved slightly. 

Response: We appreciate your valuable suggestions for the limitation and implication sections of our manuscript, which we have carefully considered and revised in the light of your suggestions and those of other reviewers. Once again, we sincerely thank you for your expertise and feedback in improving the quality of our manuscripts.

In lines 387 to 417 of the revised manuscript, the main changes are as follows:

“4.2 Limitations

Our study is not without limitations. One such limitation is that we only examined students who have already made their major choices and could not fully capture the mobility of students across different majors. Future longitudinal research could be conducted to investigate the causal relationship between gender and major choices and to explore the specific mechanisms driving these gender differences. Another limitation is that our study only focused on majors within financial and economic universities, which may limit the generalizability of our findings. To address this limitation, future research could expand the scope of the study to include comprehensive universities in order to provide a broader based evidence.”

“4.3 Practical Implications

Our findings have significant educational implications for promoting gender equality and social justice in China's higher education system, particularly, under the new broad discipline enrollment system. Females are more likely to enter popular applied majors, which can provide them with greater opportunities to alleviate their disadvantaged position in the labor market. However, it is important to acknowledge that applied majors in certain fields, such as finance and accounting, also have limitations in their employment direction. Increased numbers of female graduates in applied majors also create a risk of female employees facing intensified competitive pressures in these industries. Therefore, as long as gender stereotypes persist in the job market, gender equity issues such as opportunities and treatment should be highlighted. Meanwhile, interventions should work towards place great emphasis on guiding students in their major choices, providing them with adequate professional introductions and employment information to reduce students' lack of understanding or blind choice when selecting majors.”

Having read the introduction thus far, it is unclear how or why you choose which variables to control, and in doing so you may have omitted other one's that seam to be also verys important, such as education level of parents, scholarship opportunities or even math anxiety!

Response: We appreciate your valuable suggestions for our manuscript. In response to your question about reading this section in the PDF file that you don't understand why we control these variables, we have chosen them primarily around our research theme and to lay the groundwork for the research questions that follow our manuscript. Secondly, for the variables you mentioned about parents' education level and math score, we mentioned them in the Data and Programs section of the article. Once again, we sincerely thank you for your expertise and feedback in improving the quality of our manuscripts.

Does Ceteris paribus refers to the variables you are controling for or it to the abovementioned reforms of your educational system?

Response: Thank you for your valuable comments on our manuscript. To answer your question about reading this section of the PDF file, the other conditions in our section refer to the variables being controlled. Once again, we sincerely thank you for your expertise and feedback in improving the quality of our manuscripts.

why predominantly? is it more than 50%, more than 90% and what were other participants?

Response: Thank you for your valuable comments on our manuscript. In response to your question about reading this section of the PDF file, we have revised the presentation of this section to include all undergraduate students from universities of finance and economics. Once again, we sincerely thank you for your expertise and feedback in improving the quality of our manuscripts.

In lines 146 to 147 of the revised manuscript, the main changes are as follows:

“The participants in this study are undergraduate students studying at universities of finance and economics.”

this is considered too wide of a temporal range for most studies, specially since COVID-19 restrictions changed a lot during that period, and I speculate it could also mean differences in online vs. live classes. If so, please include that in final section regarding limitations and possible explanations.

Response: Thank you for your valuable suggestions on the limitations of our manuscript. Regarding the issues you mentioned in the PDF file, we have thought carefully and improved the content of this part. Once again, we sincerely thank you for your expertise and feedback in improving the quality of our manuscripts.

In lines 388 to 403 of the revised manuscript, the main changes are as follows:

“Our study is not without limitations. The first limitation is that we only examined students who have already made their major choices and could not fully capture the mobility of students across different majors. Future longitudinal research could be conducted to investigate the causal relationship between gender and major choices and to explore the specific mechanisms driving these gender differences. The second limitation is that our study only focused on majors within financial and economic universities, which may limit the generalizability of our findings. To address this limitation, future research could expand the scope of the study to include comprehensive universities in order to provide a broader based evidence. Finally, the time period of our study is during the COVID-19 period, and it is too long compared to the time frame of most other studies, data collection may be affected by the pandemic, and students' choice of major may also be due to changes in the format of the curriculum due to the pandemic, such as from offline to online, resulting in changes in students' attitudes towards certain majors. These can lead to less robust regression results. Future research can explore the impact of these factors on undergraduate majors in finance and economics universities without or as little impact from the pandemic.”

Reviewer 4 Report

Dear Authors,

Thanks for giving me a chance to read this manuscript, “An analysis of Factors Influencing University Students' Major Choice from the Perspective of Gender Differences”. The current paper a comprehensive approach to analyze the influence of gender differences on major choice of finance and economics university students. Moreover, this study explores the differential impact of income level, urban-rural settings, and regional differences on university students' major choices.

This is an interesting and significant topic in the field of education equity. However, there are some issues in the current manuscript that should be addressed.

1.      Method

·        The most ambiguous concern for me is the sampling technique. As the author mentioned, “

Upon 151 gathering a total of 1200 responses, the survey was concluded, ultimately obtaining 1164 152 valid questionnaires.

The survey was distributed to a total of 1,200 students. Participating universities included Shandong University of Finance and Economics, Anhui University of Finance and Economics and Jiangxi University of Finance and Economics, among others.”

For me, it is a convenience sampling, how can you ensure that 1k responses were sufficient enough for the current conclusion?

2.      Discussion

·        The discussion is much missing. There are no extensive discussion compared with other literatures, which should be seriously improved.

To sum up, I personally like this paper. However, the problems should be addressed in order to be further considered. Hope these suggestions help.

should be improved seriously

Author Response

Response to comments by Reviewer #4

Thanks for giving me a chance to read this manuscript, “An analysis of Factors Influencing University Students' Major Choice from the Perspective of Gender Differences”. The current paper a comprehensive approach to analyze the influence of gender differences on major choice of finance and economics university students. Moreover, this study explores the differential impact of income level, urban-rural settings, and regional differences on university students' major choices.

This is an interesting and significant topic in the field of education equity. However, there are some issues in the current manuscript that should be addressed.

Response: Thank you very much for reading this manuscript carefully, and we are very grateful to you for suggesting some revisions, we will consider your comments carefully and make revisions. The original comment is printed in black, followed by a reply in blue.

1.Method

The most ambiguous concern for me is the sampling technique. As the author mentioned, “

Upon 151 gathering a total of 1200 responses, the survey was concluded, ultimately obtaining 1164 152 valid questionnaires.

The survey was distributed to a total of 1,200 students. Participating universities included Shandong University of Finance and Economics, Anhui University of Finance and Economics and Jiangxi University of Finance and Economics, among others.”

For me, it is a convenience sampling, how can you ensure that 1k responses were sufficient enough for the current conclusion?

Response 1: Thank you for your questions about this part of our manuscript's methodology. We have carefully considered and responded to your questions, and once again we sincerely thank you for your guidance and expertise in improving the clarity and relevance of our research. We collect extensive and representative information in the form of questionnaires distributed online. First of all, we selected undergraduate students from finance and economics universities located in the eastern, central and western regions of China to participate in filling out the questionnaire, ensuring the breadth and depth of the questionnaire. Secondly, in general, the larger the sample, the more reliable the results. According to statistical principles, when the sample size exceeds a certain cut-off value, the population characteristics can be reliably inferred. As a rule of thumb, the responses of 1,200 students can already provide relatively reliable conclusions. Finally, we can also ensure the accuracy of the conclusions by selecting reasonable data analysis methods for processing and analysis. Based on your suggestions and those of other reviewers, we have revised the participant part of the manuscript.

In lines 146 to 155 of the revised manuscript, the main changes are as follows:

“The participants in this study are undergraduate students studying at universities of finance and economics. There are over 50 finance and economics universities in China, with a total of over 1 million undergraduate students [7]. To ensure a comprehensive representation of universities with varying academic reputations and geographical locations throughout the country, we extensively invited students from finance and economics universities situated in the Eastern, Central, and Western regions of China to participate in an online survey. The survey was distributed to a total of 1,200 students. Participating universities included Shandong University of Finance and Economics, Anhui University of Finance and Economics and Jiangxi University of Finance and Economics, among others.”

  1. National Bureau of Statistics of China, “China Education Yearbook,” People's Education Press, 2023.

  1. Discussion

The discussion is much missing. There are no extensive discussion compared with other literatures, which should be seriously improved.

Response 2: Thank you for your valuable suggestions for discussing this part of our manuscript. We have carefully considered and revised it in light of your suggestions, and once again we sincerely thank you for your guidance and expertise in improving the clarity and relevance of our research.

In lines 337 to 386 of the revised manuscript, the main changes are as follows:

“4.1. Discussion

The analysis allowed the RQs to be answered. The first RQ that aims to ascertain the relationship between gender and students' major choices within the context of broad discipline enrollment system backgrounds. Findings indicate significant gender differences in undergraduate students' major choices. This result aligns with the findings of Shao et al. (2022), who used administrative records from a prestigious medical school in China to investigate gender disparities in medical students' specialty choices. Existing research suggests that the disparities in innate capabilities between males and females, arising from biological and genetic factors, such as females being disadvantaged in abstract thinking and scientific cognition, manifest in the selection of disciplines and majors as students enter higher education. Consequently, gender differences in discipline and major choice are exogenous, objective, and unavoidably reflect pre-existing natural endowments [47]. This viewpoint has been extensively corroborated in research examining the gender impact on students' decisions to pursue science, technology, engineering, and mathematics (STEM) disciplines [22]. This study expands upon the existing literature by discussing gender disparities in major choices within the domain of universities of finance and economics.

Regarding the second RQ, the analysis confirms that the probability of female students choosing applied majors is significantly higher than that of male students, approximately 2.77 times higher. This finding is surprising as it appears to contradict some previous studies that reported no advantage for female students in entering popular advantageous majors [48]. Some studies suggest that differences in academic performance may be an important factor contributing to gender disparities in major choices [49]. In this study, even after controlling for academic performance and other potentially influencing variables, the analysis results robustly sup-port the significant tendency of female students to choose popular applied majors over male students. Shao et al pointed out various obstacles hindering female students from choosing high-paying surgical specialties when analyzing the likelihood of female students selecting such specialties. These obstacles include excessive physical demands, discrimination in recruitment, unfriendly work environments for women, and difficulties in balancing family responsibilities [49]. From the perspective of maintaining social gender segregation, both males and females, influenced by external factors and driven by internalized different values and potential social discrimination, make distinct decisions in education and major choices, tending towards the professions society expects of them [50]. In the current study, as the majors in finance and economics universities mainly comprise non-STEM disciplines such as accounting and finance, these majors are often considered more suitable for female students by society [23]. Therefore, in our research, female students tend to choose popular applied majors.

The last RQ pertains to the moderating effect of family background on the relationship between gender differences and major choices. This study finds that family background significantly influences the gender disparities in major choices. Specifically, females from economically developed regions and high-income households have a greater probability of choosing applied majors. A possible explanation is that females from economically developed regions and high-income households have easier access to resources and opportunities related to applied majors, making them more likely to choose these fields. On the other hand, in regions with relatively lower levels of economic development and income, females may face more social, economic, and cultural constraints, leading them to choose different types of majors. Additionally, female students from rural areas are more inclined to choose popular applied majors [51]. This finding is similar to a study by Yang & Sun [52], which showed that students from rural areas are more likely to choose applied majors than students from urban areas with broader employment prospects.”

  1. 22. D. Li, Y. Wang and L. Li, “Educational choice has greater effects on sex ratios of college STEM majors than has the greater male variance in general intelligence (g),” Intelligence,
  2. X. Shao and T. Wang, “Grey’s Anatomy: Gender Differences in Specialty Choice for Medical Students in China,” Sustainability, 2022, vol. 14(1). https://doi.org/10.3390/su14010230
  3. A. Lina and N. Emma, “Culture and the gender gap in choice of major: An analysis using sibling comparisons,” Journal of Economic Behavior & Organization, 2022, vol. 201, pp. 346-373. https://doi.org/10.1016/j.jebo.2022.07.026
  4. A. Kimberlee Shauman and H. Jill, “Gender, race-ethnicity and postdoctoral hiring in STEMM fields,” Social Science Research, 2023, vol. 113. https://doi.org/10.1016/j.ssresearch.2023.102854
  5. S. Stijn, W. Bart, F. Lot, R. Elisabeth, D. Eva, F. Filip-De and D. Wouter, “How interest fit relates to STEM study choice: Female students fit their choices better,” Journal of Vocational Behavior, 2021, vol. 129. https://doi.org/10.1016/j.jvb.2021.103614
  6. R. Gareth and S. Volker, “Gender-based segregation in education, jobs and earnings in South Africa,” World Development Perspectives, 2021, vol. 23. https://doi.org/10.1016/j.wdp.2021.100348
  7. Y. Zheng, C. Lin, C. Jason Chang and Y. Liao, “Who is able to choose? A meta-analysis and systematic review of the effects of family socioeconomic status on school choice,” International Journal of Educational Research, 2022, vol. 112. https://doi.org/10.1016/j.ijer.2022.101943
  8. S. Yang and M. Sun, “Family Background, Major Selection, and Income Return: An Empirical Study Based on the China General Social Survey (CGSS) Data,” Northwest Population Journal, 2022, vol. 41(02), pp. 52-66.

To sum up, I personally like this paper. However, the problems should be addressed in order to be further considered. Hope these suggestions help.

Response : Thank you for recognizing our manuscript, and thank you very much for reading our manuscript and giving us very important advice. We have carefully considered and revised the shortcomings and doubts that have been made in the light of your suggestions, and we sincerely thank you again for your guidance and expertise in improving the clarity and relevance of our research.

Reviewer 5 Report

The manuscript presents valuable research on an important phenomenon: gender differences in undergraduate major selection. The authors have generated a series of meaningful results. However, to ensure successful publication, the manuscript requires improvements. Thank you.

1. I would like to request the authors to address the following question: In your article, you mentioned, "the purpose of this study was to analyze how gender affects students' choice of applied majors versus non-applied majors... (lines 292-295)". Can we interpret this as the study primarily focusing on analyzing gender differences in undergraduate major selection? Alternatively, do these two statements imply any similarities or differences? Please provide a detailed explanation.

2. In the introduction section, it would be helpful for the authors to provide additional details regarding the factors that have been found to influence the choice of different majors by male and female students in previous research. Furthermore, it would be beneficial for the authors to explain their rationale for specifically selecting finance and economics majors as the focus of their study.

3. I suggest that the authors provide a thorough explanation in the “Introduction” section or “Data and Methods” Section for their choice of using applied and non-applied variables as dependent indicators. From the current perspective, it seems that the article’s description of this aspect might be inadequate.

4. The family income levels are currently labeled as "1, 2, 5". Please modify it to "1, 2, 3".

5. It is strongly recommended that the authors include clear explanations of the x-axis and y-axis in Figure 1.

6. In the "Conclusion, Limitations, and Policy Implications" section, the authors should incorporate two additional elements. Firstly, provide a detailed discussion of the research findings, including their underlying causes and connections to prior studies or existing theories. Secondly, present a more specific and detailed analysis of policy implications.

We kindly request the authors to carefully consider and address the aforementioned suggestions.

Author Response

Response to comments by Reviewer #5

The manuscript presents valuable research on an important phenomenon: gender differences in undergraduate major selection. The authors have generated a series of meaningful results. However, to ensure successful publication, the manuscript requires improvements. Thank you.

Response: Thank you for your feedback on our manuscript. We appreciate your acknowledgment of the valuable research and meaningful results presented in our study on gender differences in undergraduate major selection. We understand the importance of making improvements to ensure successful publication. We will carefully consider your comments and make the necessary revisions. Thank you again for your valuable input.

  1. I would like to request the authors to address the following question: In your article, you mentioned, "the purpose of this study was to analyze how gender affects students' choice of applied majors versus non-applied majors... (lines 292-295)". Can we interpret this as the study primarily focusing on analyzing gender differences in undergraduate major selection? Alternatively, do these two statements imply any similarities or differences? Please provide a detailed explanation.

Response 1: Thank you for your attention this. As the original article states "The purpose of this study is to analyze how gender affects students' choice of applied and non-applied majors. This includes the extent to which gender differences influence the choice of major and whether this influence varies by family background and region." We emphasize the impact of gender on students' choice of major and classify majors into two categories: applied and non-applied majors. It looks at gender differences and preferences between the two types of majors. It explores the extent to which and why men and women choose applied and non-applied majors. While you mentioned that "the study focused primarily on analyzing gender differences in undergraduate major choices," I believe that this statement puts the focus on gender differences in undergraduate major choices. Emphasizing the influence of gender in the undergraduate population on the choice of major and emphasizing that gender is an influential factor in undergraduate students' choice of major.

  1. In the introduction section, it would be helpful for the authors to provide additional details regarding the factors that have been found to influence the choice of different majors by male and female students in previous research. Furthermore, it would be beneficial for the authors to explain their rationale for specifically selecting finance and economics majors as the focus of their study.

Response 2: Thank you for your valuable feedback on our manuscript. We appreciate your suggestion to provide additional details in the introduction section regarding the factors that have been found to influence the choice of different majors by male and female students in previous research. We will include a more comprehensive discussion in this area to enhance the context of our study. Furthermore, we understand the importance of explaining our rationale for choosing finance and economics majors as the focus of our study. We will provide a clear explanation in the revised manuscript to better justify our selection. Thank you again for your insightful comments. We will make the necessary revisions to address these points. In response to the presented question, we have provided the following content for clarification.

In lines 60 to 74 of the revised manuscript, the main changes are as follows:

“Scholars argue that societal expectations of different roles and expectations for males and females in specific industries may influence the major choices of male and female students [23]. Engineering, computer science, and other STEM fields are considered more suitable for males, while fields such as education, sociology, and economics are considered more suitable for females. These notions to some extent influence students' level of interest and willingness in pursuing certain majors [23]. Additionally, educational background and family environment can also have an impact on the major choices of male and female students [24]. Schools and families often convey specific gender role ideologies through subtle guidance and education, which in turn affect students' major choices [24]. For instance, parents' educational levels serve as a factor; when parents have lower levels of education, they may not consider all aspects when assisting their children in selecting a major [25]. Conversely, when parents have higher levels of education, they are often able to provide valuable advice by analyzing factors such as academic difficulty, major prospects, salary potential, and societal value when guiding their children in choosing a major [25].”

  1. X. Shao and T. Wang, “Grey’s Anatomy: Gender Differences in Specialty Choice for Medical Students in China,” Sustainability, 2022, vol. 14(1). https://doi.org/10.3390/su14010230
  2. H. Tao and H. Zheng, “Parental and sibling influence on study field choice: Gender-stereotypical or field preference transmission,” Journal of Asian Economics, 2022, vol.82. https://doi.org/10.1016/j.asieco.2022.101509
  3. Y. Sun, X. Zhang and Q. Ding, “The Influence of Parental Occupational Expectations on Children's College Entrance Examination Voluntary Choices,” New Curriculum Teaching (Electronic), 2020, vol. 108(24), pp. 98-99.
  4. I suggest that the authors provide a thorough explanation in the “Introduction” section or “Data and Methods” Section for their choice of using applied and non-applied variables as dependent indicators. From the current perspective, it seems that the article’s description of this aspect might be inadequate.

Response 3: Thank you for your comment. Based on your feedback, we have added a separate section in 2. Data and Methods to explain the applied and non-applied disciplines.

In lines 164 to 181 of the revised manuscript, the main changes are as follows:

“2.3. Measures

The dependent variable in our study was the decision of undergraduate students to enroll in an applied or non-applied major at universities of finance and economics. This variable was defined as a binary variable, where a value of 1 indicated a student choosing an applied major and a value of 0 indicated a student not choosing an ap-plied major. Based on the definition of Beecher et al. [37], we consider majors that are vocational, technical, and involve specific scenarios as applied majors. Applied disciplines are characterized by strong practicality and skill requirements, higher alignment with the market, better fitting of enterprise skill requirements, adaptability to complex work needs, and more lucrative returns in the labor market [38]. These majors include Accounting, Financial Management, Auditing, International Business, Marketing, Human Resource Management, Logistics Management, E-commerce, Engineering Cost, Computer Science and Technology, Business English, and others. Non-applied majors include Economics, National Economic Management, Public Finance, Journal-ism, Statistics, Taxation, Trade Economics, Finance, Business Administration, Mathematics, and others. From the survey results, it was found that students choosing non-applied majors slightly outnumbered those choosing applied majors. Of the 1,162 valid questionnaires, 538 students (46.22%) chose an applied major, while 626 students (53.78%) chose a non-applied major.”

  1. Y. Tang, M. Pu and H. Chen, “T. Beecher and Trollell: Academic tribes and their territories: knowledge exploration and disciplinary culture (retranslation),” Peking University Press, 2015, pp. 40-41.
  2. The family income levels are currently labeled as "1, 2, 5". Please modify it to "1, 2, 3".

Response 4: Thank you for bringing this to our attention. We apologize for the oversight in labeling the family income levels as “1, 2, 5” instead of “1, 2, 3”. We will make the necessary correction in the revised version of the manuscript. Your feedback is greatly appreciated, and we are committed to improving the accuracy and clarity of our research.

In the revised manuscript, the main changes are as follows. We changed the numbers in the Table 1. The revised sentence reads:

“Annual household income level (1 = less than 150,000; 2 = 15-450,000; 3 = 450,000 or more)”

  1. It is strongly recommended that the authors include clear explanations of the x-axis and y-axis in Figure 1.

Response 5: We appreciate your suggestion to include clear explanations of the x-axis and y-axis in Figure 1. We understand the importance of providing a comprehensive understanding of the figure for the readers. In the revised version of the manuscript, we will ensure that the x-axis and y-axis are properly labeled with clear explanations to enhance the clarity of Figure 1. Thank you for highlighting this, and we will make the necessary revisions to address this suggestion.

In lines 287 to 288 of the revised manuscript, we added a note of the x- and y-axis. The revised sentence reads:

“Note:the x-axis represents gender and the y-axis represents the degree to which a student's family background is correlated with his or her major choices.”

  1. In the "Conclusion, Limitations, and Policy Implications" section, the authors should incorporate two additional elements. Firstly, provide a detailed discussion of the research findings, including their underlying causes and connections to prior studies or existing theories. Secondly, present a more specific and detailed analysis of policy implications.

Response 6: Thank you for your suggestion. Based on your feedback and that of other reviewers, we have made appropriate revisions to the "Conclusion, Limitations, and Policy Implications" section. We have included a detailed discussion of the research findings, including their underlying causes and connections to prior studies or existing theories. Additionally, we have provided a more specific and detailed analysis of the policy implications. These modifications will enhance the quality and comprehensibility of the paper. Thank you again for your valuable input!

In lines 337 to 438 of the revised manuscript, the main changes are as follows:

“4.1. Discussion

The analysis allowed the RQs to be answered. The first RQ that aims to ascertain the relationship between gender and students' major choices within the context of broad discipline enrollment system backgrounds. Findings indicate significant gender differences in undergraduate students' major choices. This result aligns with the findings of Shao et al. (2022), who used administrative records from a prestigious medical school in China to investigate gender disparities in medical students' specialty choices. Existing research suggests that the disparities in innate capabilities between males and females, arising from biological and genetic factors, such as females being disadvantaged in abstract thinking and scientific cognition, manifest in the selection of disciplines and majors as students enter higher education. Consequently, gender differences in discipline and major choice are exogenous, objective, and unavoidably reflect pre-existing natural endowments [47]. This viewpoint has been extensively corroborated in research examining the gender impact on students' decisions to pursue science, technology, engineering, and mathematics (STEM) disciplines [22]. This study expands up-on the existing literature by discussing gender disparities in major choices within the domain of universities of finance and economics.

Regarding the second RQ, the analysis confirms that the probability of female students choosing applied majors is significantly higher than that of male students, ap-proximately 2.77 times higher. This finding is surprising as it appears to contradict some previous studies that reported no advantage for female students in entering popular advantageous majors [48]. Some studies suggest that differences in academic performance may be an important factor contributing to gender disparities in major choices [49]. In this study, even after controlling for academic performance and other potentially influencing variables, the analysis results robustly sup-port the significant tendency of female students to choose popular applied majors over male students. Shao et al pointed out various obstacles hindering female students from choosing high-paying surgical specialties when analyzing the likelihood of female students selecting such specialties. These obstacles include excessive physical demands, discrimination in recruitment, unfriendly work environments for women, and difficulties in balancing family responsibilities [49]. From the perspective of maintaining social gender segregation, both males and females, influenced by external factors and driven by internalized different values and potential social discrimination, make distinct decisions in education and major choices, tending towards the professions society expects of them [50]. In the current study, as the majors in finance and economics universities mainly comprise non-STEM disciplines such as accounting and finance, these majors are often considered more suitable for female students by society [23]. Therefore, in our research, female students tend to choose popular applied majors.

The last RQ pertains to the moderating effect of family background on the relationship between gender differences and major choices. This study finds that family background significantly influences the gender disparities in major choices. Specifically, females from economically developed regions and high-income households have a greater probability of choosing applied majors. A possible explanation is that females from economically developed regions and high-income households have easier access to resources and opportunities related to applied majors, making them more likely to choose these fields. On the other hand, in regions with relatively lower levels of eco-nomic development and income, females may face more social, economic, and cultural constraints, leading them to choose different types of majors. Additionally, female students from rural areas are more inclined to choose popular applied majors [51]. This finding is similar to a study by Yang & Sun [52], which showed that students from rural areas are more likely to choose applied majors than students from urban areas with broader employment prospects.

4.2 Limitations

Our study is not without limitations. The first limitation is that we only examined students who have already made their major choices and could not fully capture the mobility of students across different majors. Future longitudinal research could be conducted to investigate the causal relationship between gender and major choices and to explore the specific mechanisms driving these gender differences. The second limitation is that our study only focused on majors within financial and economic universities, which may limit the generalizability of our findings. To address this limitation, future research could expand the scope of the study to include comprehensive universities in order to provide a broader based evidence. Finally, the time period of our study is during the COVID-19 period, and it is too long compared to the time frame of most other studies, data collection may be affected by the pandemic, and students' choice of major may also be due to changes in the format of the curriculum due to the pandemic, such as from offline to online, resulting in changes in students' attitudes towards certain majors. These can lead to less robust regression results. Future research can explore the impact of these factors on undergraduate majors in finance and eco-nomics universities without or as little impact from the pandemic.

4.3 Practical Implications

Our findings have significant educational implications for promoting gender equality and social justice in China's higher education system, particularly, under the new broad discipline enrollment system. Females are more likely to enter popular ap-plied majors, which can provide them with greater opportunities to alleviate their dis-advantaged position in the labor market. However, it is important to acknowledge that applied majors in certain fields, such as finance and accounting, also have limitations in their employment direction. Increased numbers of female graduates in applied majors also create a risk of female employees facing intensified competitive pressures in these industries. Therefore, as long as gender stereotypes persist in the job market, gen-der equity issues such as opportunities and treatment should be highlighted. Mean-while, interventions should work towards place great emphasis on guiding students in their major choices, providing them with adequate professional introductions and employment information to reduce students' lack of understanding or blind choice when selecting majors.

4.4 Conclusion

The literature on the major choices in higher education in China has mostly focused on the traditional enrollment model, therefore, there is little research analyzing the impact of gender on major choices under the broad discipline enrollment system. Against the backdrop of the emerging and widely promoted broad discipline enrollment reform in China in recent years, our research has advanced the existing literature. Specifically, the purpose of this study was to analyze how gender affects students’ choice of applied majors versus non-applied majors. This includes the degree to which gender differences have an impact on professional choices, and whether this impact will vary depending on family background and region. In recent research, gender classifications have become increasingly diverse. Due to our inability to analyze gender diversity, we follow Cui et al. [53] in categorizing gender mainly as male and female. By using the LDM model to analyze who was more likely to enter advantaged applied majors in universities, we focus on gender differences in professional education re-sources during the undergraduate stage. Furthermore, we explored the differential im-pact of gender on major choices due to differences in family background (such as in-come level, urban and rural areas, and regions). Our study provided empirical evidence that, under the broad discipline enrollment system, the probability of female students choosing applied majors is significantly higher than that of male students and this gap increases with the increase of family income levels. Furthermore, female students from Eastern regions and rural areas in China are more likely to choose applied majors than females in other areas.”

  1. 22. D. Li, Y. Wang and L. Li, “Educational choice has greater effects on sex ratios of college STEM majors than has the greater male variance in general intelligence (g),” Intelligence,
  2. X. Shao and T. Wang, “Grey’s Anatomy: Gender Differences in Specialty Choice for Medical Students in China,” Sustainability, 2022, vol. 14(1). https://doi.org/10.3390/su14010230
  3. A. Lina and N. Emma, “Culture and the gender gap in choice of major: An analysis using sibling comparisons,” Journal of Economic Behavior & Organization, 2022, vol. 201, pp. 346-373. https://doi.org/10.1016/j.jebo.2022.07.026
  4. A. Kimberlee Shauman and H. Jill, “Gender, race-ethnicity and postdoctoral hiring in STEMM fields,” Social Science Research, 2023, vol. 113. https://doi.org/10.1016/j.ssresearch.2023.102854
  5. S. Stijn, W. Bart, F. Lot, R. Elisabeth, D. Eva, F. Filip-De and D. Wouter, “How interest fit relates to STEM study choice: Female students fit their choices better,” Journal of Vocational Behavior, 2021, vol. 129. https://doi.org/10.1016/j.jvb.2021.103614
  6. R. Gareth and S. Volker, “Gender-based segregation in education, jobs and earnings in South Africa,” World Development Perspectives, 2021, vol. 23. https://doi.org/10.1016/j.wdp.2021.100348
  7. Y. Zheng, C. Lin, C. Jason Chang and Y. Liao, “Who is able to choose? A meta-analysis and systematic review of the effects of family socioeconomic status on school choice,” International Journal of Educational Research, 2022, vol. 112. https://doi.org/10.1016/j.ijer.2022.101943
  8. S. Yang and M. Sun, “Family Background, Major Selection, and Income Return: An Empirical Study Based on the China General Social Survey (CGSS) Data,” Northwest Population Journal, 2022, vol. 41(02), pp. 52-66.
  9. N. Cui, R. Wang, F. Song and J. Jin, “Experiences and Perceptions of Male Nursing Students in a Single-Sex Class: A Qualitative Descriptive Study,” Nurse Education in Practice, 2021, vol. 51. https://doi.org/10.1016/j.nepr.2021.102996

Round 2

Reviewer 1 Report

Thanks for opportunity review revised manuscript entitled ‘‘An analysis of Factors Influencing Chinese University Students' Major Choice from the Perspective of Gender Differences’’. I would like the thanks to authors. They make a good job for improving quality of their manuscript. Authors revised the manuscript as I requested with a good will. In this form, Introduction reflects very well the previous studies and study aim, Method section and Result section is correct, and Discussion section adequately synthesis to previous study findings and current study results. Overall, I have no further comment regarding to manuscript. I congratulate to authors and wish them success on their future endeavors.

Minor English editing required. 

Reviewer 2 Report

Dear author/s, 

I was glad to see that your manuscript has been revised and consequently it could be published in this present form.

Reviewer 4 Report

The authors have addressed most of my concerns. I am happy to recommended it for publication.

The authors have addressed most of my concerns. I am happy to recommended it for publication.